# Phosphoregulation of HORMA domain protein HIM-3 promotes asymmetric synaptonemal complex disassembly in meiotic prophase in *Caenorhabditis elegans*

**Aya Sato-Carlton**[1], **Chihiro Nakamura-Tabuchi**[1¤a], **Xuan Li**[1], **Hendrik Boog**[1¤b], **Madison K. Lehmer**[2¤c], **Scott C. Rosenberg**[2¤d], **Consuelo Barroso**[3], **Enrique Martinez-Perez**[3], **Kevin D. Corbett**[2,4,5], **Peter Mark Carlton**[1,6,7] *

**1** Kyoto University, Graduate School of Biostudies, Japan, **2** Department of Chemistry and Biochemistry, University of California, San Diego, United States of America, **3** MRC London Institute of Medical Sciences, Imperial College, London, **4** Department of Cellular and Molecular Medicine, University of California, San Diego, United States of America, **5** Ludwig Institute for Cancer Research, San Diego Branch, United States of America, **6** Kyoto University, Radiation Biology Center, Japan, **7** Institute for Integrated Cell-Material Sciences (iCeMS), Kyoto University, Japan

¤a Current address: Department of Nutritional Sciences and Toxicology, University of California, Berkeley, CA, United States of America

¤b Current address: Institut für Pharmazie und Molekulare Biotechnologie (IPMB), Ruprecht-Karls-Universität Heidelberg, Heidelberg, DE, Germany

¤c Current address: Molecular and Cell Biology Graduate Program, University of California Berkeley, Berkeley CA, United States of America

¤d Current address: Genentech, Inc., South San Francisco, CA, United States of America

* carlton.petermark.3v@kyoto-u.ac.jp

## Abstract

In the two cell divisions of meiosis, diploid genomes are reduced into complementary haploid sets through the discrete, two-step removal of chromosome cohesion, a task carried out in most eukaryotes by protecting cohesion at the centromere until the second division. In eukaryotes without defined centromeres, however, alternative strategies have been innovated. The best-understood of these is found in the nematode *Caenorhabditis elegans*: after the single off-center crossover divides the chromosome into two segments, or arms, several chromosome-associated proteins or post-translational modifications become specifically partitioned to either the shorter or longer arm, where they promote the correct timing of cohesion loss through as-yet unknown mechanisms. Here, we investigate the meiotic axis HORMA-domain protein HIM-3 and show that it becomes phosphorylated at its C-terminus, within the conserved "closure motif" region bound by the related HORMA-domain proteins HTP-1 and HTP-2. Binding of HTP-2 is abrogated by phosphorylation of the closure motif in *in vitro* assays, strongly suggesting that *in vivo* phosphorylation of HIM-3 likely modulates the hierarchical structure of the chromosome axis. Phosphorylation of HIM-3 only occurs on synapsed chromosomes, and similarly to other previously-described phosphorylated proteins of the synaptonemal complex, becomes restricted to the short arm after designation of crossover sites. Regulation of HIM-3 phosphorylation status is required for timely disassembly of synaptonemal complex central elements from the long arm, and is also required for

**Data Availability Statement:** All relevant data are within the manuscript and its Supporting Information files.

**Funding:** This work was supported by a Japan Society for the Promotion of Science RPD fellowship and Naito foundation grant for female scientists to A. S-C., Japan Society for the Promotion of Science KAKENHI grants (24687024 Wakate A and 15H04328 Kiban B to P.M.C, and 17K15064 Wakate B and19K06486 Kiban C to A.S-C.), Kyoto university SPIRITS grant to P.M.C., National Institutes of Health grant number R01-GM104141 to K.D.C., and MRC grant MC-A652-5PY60 to E.M-P. The funders had no role in study design, data collection and analysis, decision to publish, or preparation of the manuscript.

**Competing interests:** The authors have declared that no competing interests exist.

proper timing of HTP-1 and HTP-2 dissociation from the short arm. Phosphorylation of HIM-3 thus plays a role in establishing the identity of short and long arms, thereby contributing to the robustness of the two-step chromosome segregation.

## Author summary

To segregate properly in meiosis, cohesion between replicated chromosomes must remain after the first meiotic cell division, so chromosomes can be held together until they finally separate in the second division. While the majority of organisms use centromeres to protect chromosome cohesion in the first division, the nematode worm *C. elegans*, which lacks single centromeres, instead protects cohesion only on a segment of the chromosome known as the "long arm". The long arm (and its complement, the short arm) are known to accumulate specific proteins and protein modifications, but it is not known how the short and long arms are first distinguished, nor how their separate functions are carried out. We report here that the chromosome axis protein HIM-3 and its modification by phosphorylation is important for ensuring the robust establishment of short and long arm functions. We show that phosphorylated HIM-3 partitions to the short arms after crossover recombination sites are designated, and HIM-3 mutants that mimic constitutive phosphorylation delay the normal establishment of the two complementary arm domains. Our findings reveal another layer of regulation to an outstanding mystery in chromosome biology.

## Introduction

Meiosis reduces chromosome number from diploid to haploid by carrying out two rounds of chromosome segregation following a single round of DNA replication. To segregate chromosomes correctly over the course of two divisions, two distinct chromosomal domains must be established as sites of cohesion loss for the first and second meiotic divisions, where cohesin complexes joining sister chromatids at each site are successively degraded. Organisms whose chromosomes have single, defined centromeres (monocentric) lose cohesin from chromosome arms in meiosis I, while cohesin at centromeres is protected by the protein Shugoshin [reviewed in 1] until meiosis II. In contrast, organisms with holocentric chromosomes, such as *Caenorhabditis elegans*, must define these two domains using a different mechanism. In *C. elegans*, each chromosome normally receives a single crossover (CO) [reviewed in 2], which divides the chromosome into two domains of unequal length termed the short and long arms [3,4]. At meiosis I, cohesion is removed specifically from short arms, while cohesion on long arms persists until degraded at meiosis II, thus separating chromatids in two discrete steps [reviewed in 5]. Although crossover position is biased towards the terminal thirds of chromosomes [6], crossovers can potentially occur at any position along the chromosome. Therefore, establishment of short and long arm domains must occur facultatively for each chromosome in each meiocyte.

The synaptonemal complex (SC) is a protein macroassembly that plays a critical role in holding homologous chromosomes together to ensure formation of crossovers during meiotic prophase [reviewed in 7]. Once crossovers are formed, the SC is disassembled in a stepwise manner, which influences the timing of cohesin degradation during the two subsequent meiotic divisions. In both monocentric and holocentric organisms, SC components disassemble

in a non-uniform manner [3,8–13]. SC components are actively maintained at paired centromeres, and disassemble from chromosome arms, in diplotene and diakinesis in budding yeast, flies and mice, while in contrast, specific SC components are maintained either at short or long arms after SC disassembly in worms. This stepwise or asymmetric disassembly of SC components is suggested to promote centromere bi-orientation and kinetochore function, and establish chromosome separation domains [14]. However, the mechanisms regulating asymmetric SC disassembly are poorly understood.

In *C. elegans*, multiple upstream factors interacting with the SC, as well as post-translational regulation of SC components, have been found to be important for chromosome partitioning into short and long arms. The E3 ligases ZHP-1 and ZHP-2, which help ensure the formation of a single crossover per chromosome, localize to short arms and are required for asymmetric SC disassembly [15]. Another pair of E3 ligases, ZHP-3 and ZHP-4, which localize to crossovers, are also important for asymmetric disassembly [16–18]. In addition, phosphorylation of the SC transverse and central elements SYP-1 and SYP-2 is known to promote establishment of short and long arms [19,20]. MAP kinase-mediated phosphorylation of SYP-2 and subsequent inactivation of MAPK upon crossover designation triggers disassembly of SC central element proteins from the long arm [19]. Phosphorylation at the Polo Box Domain (PBD) binding motif of SYP-1 at Thr452 promotes the cascade of short and long arm establishment by recruiting PLK-2 to the SC. Phosphorylated SYP-1 and PLK-2 are mutually dependent for their colocalization on short arms upon crossover designation, and a non-phosphorylatable *syp-1* mutation leads to loss of asymmetry and failure in meiosis I segregation [20]. Once short/long asymmetry is established, the chromosome passenger complex (CPC) containing INCENP[ICP-1] and Aurora B[AIR-2] is recruited to the short arm by Haspin kinase-mediated phosphorylation of histone H3Thr3 [20–23]. H3Thr3 in turn is dephosphorylated on the long arm by HTP-1/LAB-1-based recruitment of phosphatase PP1[GSP-2] [23,24]. AuroraB[AIR-2] then phosphorylates cohesin REC-8 on the short arm, allowing its removal at meiosis I [21–23]. Taken together, these studies have shown that asymmetric partitioning of SC components and their interactors collectively ensures the establishment of chromosome domains required for correct segregation. However, the mechanisms that sense the crossover intermediates, partition chromosomes in a length-sensitive manner and disassemble SC components in an asymmetric manner are not understood well.

The HORMA domain family proteins (HORMADs) were first identified by sequence conservation among Hop1, Rev7 and Mad2 in budding yeast, and act in cellular processes such as meiosis, mitotic cell cycle control and the spindle assembly checkpoint [25,reviewed in 26]. Previous studies have shown that the HORMA domain contains a C-terminal region termed a "safety belt" which wraps around, and locks in place, the "closure motif" coil of a binding partner [27].There are four HORMA domain proteins that make up the meiotic chromosome axis in *C. elegans*: the paralogs HTP-1 and HTP-2, which share 82% sequence identity and have nearly identical HORMA domain structures, HTP-3, and HIM-3 [27–31]. Each protein contains an N-terminal HORMA domain and at least one C-terminal closure motif [27]. With HTP-3 as the platform of assembly, these four proteins assemble in a hierarchical manner to form the axis, in which HTP-1/2 bind to HIM-3's sole closure motif as well as to HTP-3's closure motifs 1 and 6, while HIM-3 binds to HTP-3's closure motifs 2, 3, 4 and 5 [27]. Once crossover intermediates are formed, HTP-1/2 are removed from short arms and persist on long arms to recruit LAB-1 and PP1[GSP-2] for protection of cohesion on long arms, while HIM-3 and HTP-3 remain on both short and long arms [4,23,24,32]. The molecular mechanisms removing HTP-1/2 from short arms, and SC central element proteins from long arms, in response to crossover designation remains a mystery.

Here, we characterize phosphorylation sites within the closure motif of HIM-3, and find that this phosphorylation promotes asymmetric SC disassembly as meiocytes exit pachytene in *C. elegans*. Phosphorylated HIM-3 localizes along the entire length of the SC in early meiotic prophase, and later becomes enriched at short arms once crossover intermediates are formed. We found that under conditions with limited crossover formation, this asymmetric partitioning of phosphorylated HIM-3 fails when fewer than four crossovers are present, suggesting that a dosage-sensitive mechanism triggers chromosome partitioning. Mutations in *him-3* phosphoresidues also delay asymmetric disassembly of the SC upon the exit from pachytene, suggesting that phosphoregulation of HIM-3 is part of the multilayered process establishing chromosome arm identity and thus ensuring meiotic chromosome segregation.

## Results

### HIM-3$^{HORMAD1/2}$ is phosphorylated at the conserved closure motif

To understand regulation of meiotic proteins by post-translational modification, we previously carried out mass spectrometry to identify phosphorylated proteins in *C. elegans* adult hermaphrodites [20]. In two independent experiments, we found that the chromosome axis protein HIM-3 is phosphorylated at its C-terminus. We detected phosphopeptides containing Ser282 either singly or in combination with Ser274, Ser277, or Tyr279, and the most commonly-observed peptide contained doubly-phosphorylated Ser277/Ser282 (**Fig 1A** and **S1 Table**). Among the three most C-terminal phosphoresidues, which reside within the conserved closure motif, Tyr279 and Ser or Thr at position 282 are conserved features among nematode HIM-3 orthologs (**Fig 1A**). Mouse HORMAD1 Ser375 was shown to be phosphorylated on unsynapsed chromosomes, and this phosphorylation is hypothesized to signal the status of synapsis [33]. More recent studies further identified this C-terminus region containing Ser375 to be the closure motif, able to be bound by the HORMA domain of HORMAD1 itself and HORMAD2 [27,34]. While phosphorylation at the closure motif is conserved between *C. elegans* HIM-3 and mouse HORMAD1 (**S1 Fig**), the function is not understood. Since other HORMA family proteins, HTP-1 and -2, are known to bind the HIM-3 closure motif in *C. elegans* [27], we first examined if phosphorylation affects HTP-1/2 binding to HIM-3 *in vitro*. A previous study has shown that both HTP-1 and -2 proteins bind unmodified HIM-3 peptides containing the closure motif in a fluorescence polarization (FP) peptide binding assay [27]. Examination of the crystal structure resolved by the same study [27] led us to predict that phosphorylation at Ser282 would likely interfere with HTP-1/2 binding (**Fig 1B**). We conducted a similar FP peptide binding assay using purified HTP-2 and fluorescently tagged HIM-3 closure motif peptides containing various modifications (**Fig 1C**). As expected, HTP-2 proteins bound to the unmodified HIM-3 peptide (*Kd* = 5.8μM). In contrast, phosphorylation at Ser282 either solely or in combination with phosphorylation of other residues completely abrogated HTP-2 binding to HIM-3 closure motif peptides (*Kd*: no significant binding detected), while phosphorylation at HIM-3 Tyr279 mildly but significantly lowered binding (*Kd* = 16.1μM, p value <0.0001 by unpaired t-test). This suggested the possibility that HTP-1/2 binding to HIM-3 is modulated by HIM-3 phosphorylation status at the closure motif.

### Phosphorylated HIM-3 localizes to the entire length of the SC and then becomes enriched on the short arms

To further understand the function of HIM-3 phosphorylation, we generated phospho-specific antibodies against HIM-3 peptides containing phosphorylated Ser277 and Ser282 since we detected this doubly-phosphorylated peptide most frequently by mass spectrometry. Lack of

A

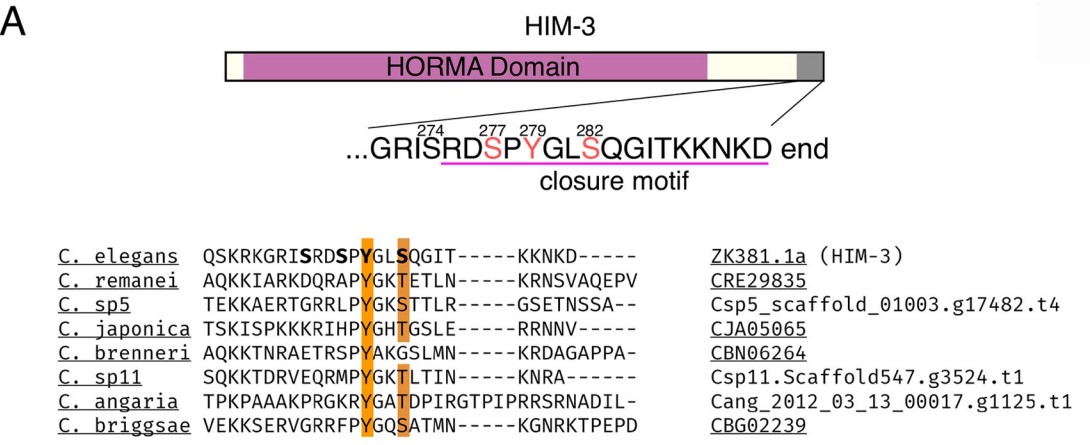

B

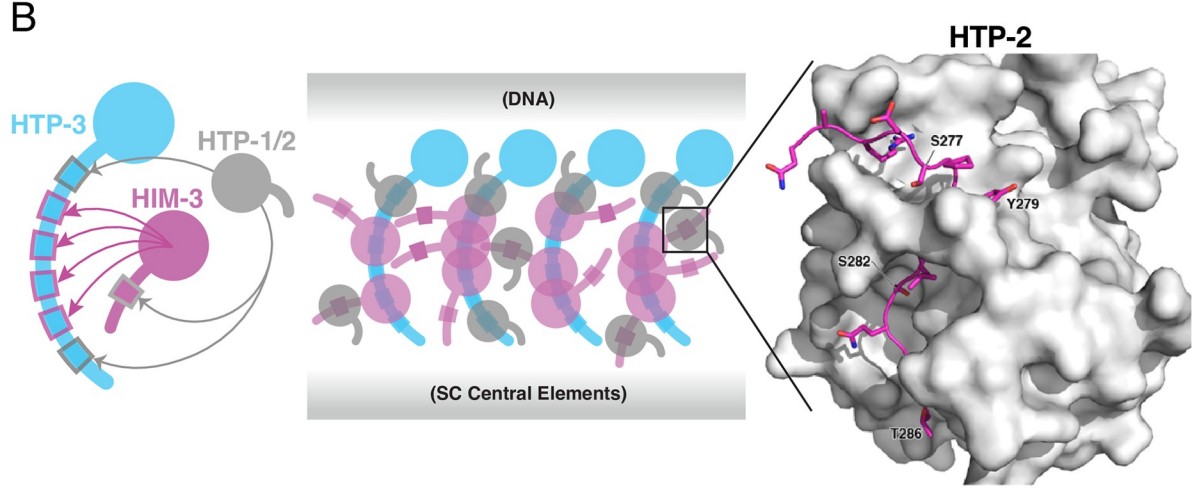

C

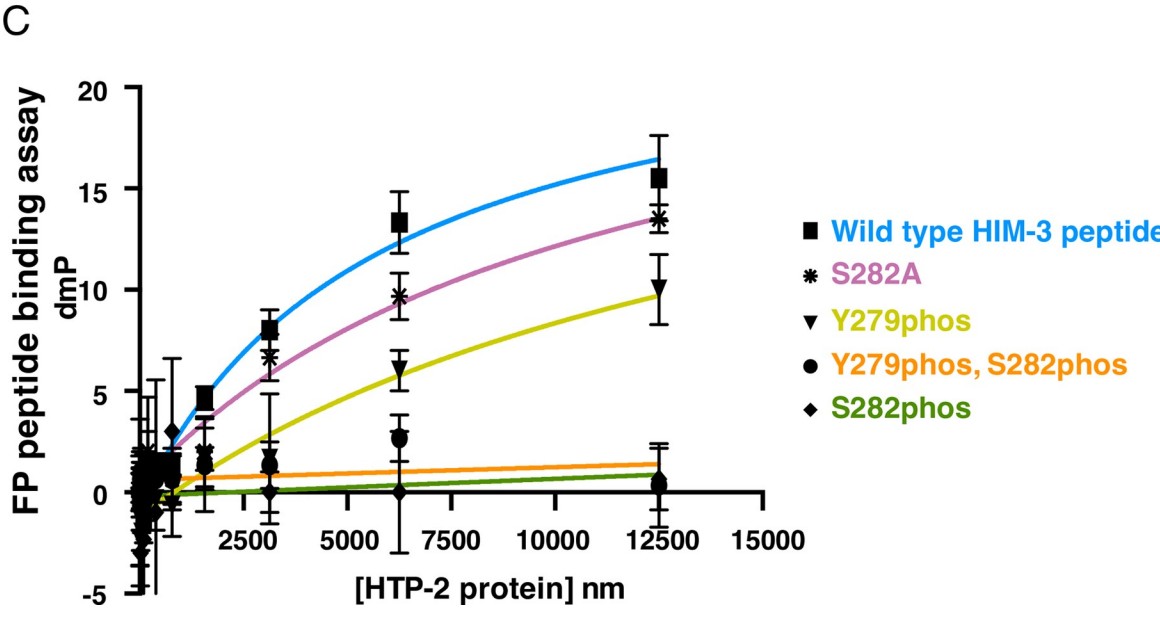

**Fig 1. Phosphorylation at the HIM-3 closure motif prevents HTP-1/2 binding *in vitro*. A**, *top*: Schematic diagram of HIM-3, showing the HORMA domain (magenta) and phosphorylation sites identified by mass spectrometry in red. The conserved closure motif is underlined. *bottom*: Conservation of the Tyr and Ser/Thr, highlighted in orange, in the closure motif in eight *Caenorhabditis* species. **B**, *left*: Diagram of hierarchical binding of HTP-1, 2, 3 and HIM-3 from [27]. *center*, diagram of possible multivalent binding modes of the axial element, based on [36,71]. *right*, Interactions between the HIM-3 closure motif (magenta) and the HTP-2 (grey) HORMA domain shown in PDB structure 4TZL [27]. **C**, Fluorescence polarization peptide binding assay using bacterially purified HTP-2 protein (full length) and fluorescently tagged HIM-3 peptides with indicated phosphorylation or Ala substitution. Wild type means HIM-3 peptides with no phosphorylation. Phosphorylation at Ser282 inhibits HTP-2 binding *in vitro*. Error bars are calculated as the SD from triplicated measurements. Measured Kd values were: 5.8μM for unphosphorylated peptides, not significant binding for S282phos and Y279phos_S282phos peptides, 16.1μM for Y279phos peptides, and 11.3μM for S282A peptides. The binding constants of all the modified HIM-3 peptides are statistically different from unmodified, wild type HIM-3 peptides (two-tailed p value < 0.0001 by unpaired t-test).

staining with this antibody in *him-3* mutants in which both Ser277 and Ser282 are converted to Ala confirmed specificity for the phosphoresidues (**S1 Fig**). Immunofluorescence using HIM-3phos specific antibodies showed that phosphorylation of HIM-3 starts from the beginning of meiotic prophase. From leptotene through mid-pachytene, HIM-3phos antibody staining coincides with that of "panHIM-3" antibodies that were raised against the entire protein and are thus expected to recognize both phosphorylated and unphosphorylated HIM-3 (**Fig 2A**). As oocyte precursor cells progress into late pachytene, designate crossovers, and begin to differentiate long and short arms, HIM-3phos antibody staining becomes enriched on only a small, terminal part of the SC, in contrast to panHIM-3 staining which remains on the full length of the SC, suggesting that phosphorylated HIM-3 becomes enriched on the short arm (**Fig 2B**). Simultaneous staining with the short arm marker SYP-1phos antibody [20] as well as the crossover designation marker COSA-1 [35] revealed that phosphorylated HIM-3 becomes enriched at short arms once crossovers are designated (**Fig 2C**).

Compared to the complete confinement of phosphorylated SYP-1 to short arms immediately after meiocytes enter late pachytene, phosphorylated HIM-3 enrichment was less complete and began slightly later (**Fig 3A**). Enrichment of phosphorylated HIM-3 was detected at the very end of late pachytene (typically the last 5 rows of meiocytes in late pachytene), prior to dissociation of bulk of SYP-1 from long arms, and weak HIM-3phos staining was still detectable on long arms until phosphorylated HIM-3 became completely confined to short arms at -1 diakinesis, the most mature oocyte precursor cell (proceeding distally from the spermatheca, oocyte precursors in diakinesis are designated as stage −1, −2, −3 etc. oocytes)(**Fig 2B, S1 Fig**).

Since our HIM-3phos antibody was raised against peptides carrying two phosphorylated residues (Ser277phos and Ser282phos), we further examined the specificity of this antibody by staining *him-3(S277A)* or *him-3(S282A)* single mutants (**S2 and S3 Figs**). HIM-3phos antibody stained the SC in both *him-3(S277A)* and *him-3(S282A)* single mutants, indicating that this polyclonal antibody is a mixture of antibodies recognizing Ser282phos (HIM-3phos antibody staining in S277A mutants) and antibodies recognizing Ser277phos (antibody staining in S282A mutants). While both Ser277phos and Ser282phos staining started from the transition zone along the entire length of the SC axis, Ser282phos staining (HIM-3phos staining in S277A mutants) was sparse and spotty on the SC at first, becoming more robust in late pachytene. In contrast, Ser277phos staining appeared more linear from the transition zone onward (**S2 and S3 Figs**). This disparity strongly suggests that Ser277 and Ser282 phosphorylation occurs independently. In *him-3(S277A)* mutants, the long/short arm differentiation was established normally, and Ser282phos signals became confined to short arms in late pachytene (**S2 Fig**). In fact, S282phos signals are more confined than the combined S277+S282phos signals in the wild type, where a minor fraction of HIM-3phos staining persists faintly on long arms until -2 diakinesis (**S1 and S2 Figs**). These observations suggest that Ser282phos and the bulk

A

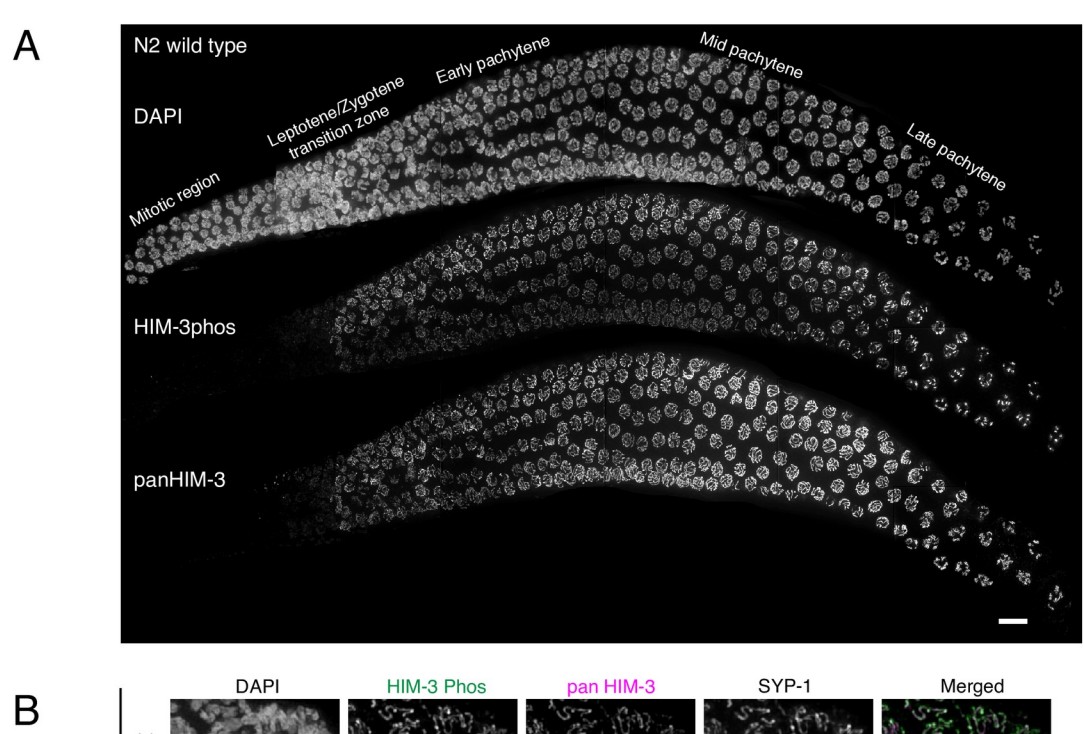

B

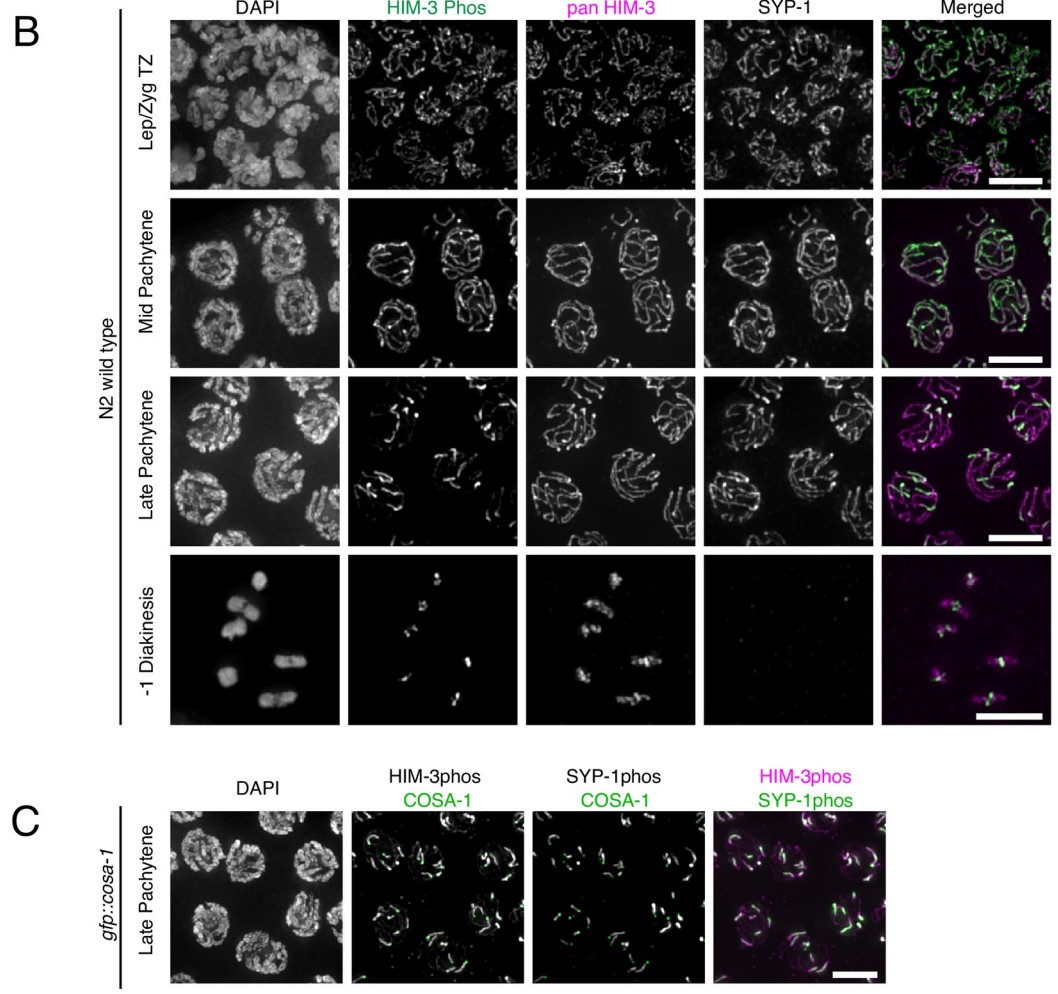

C

**Fig 2. Phosphorylated HIM-3 localizes to the SC and becomes enriched on short arms. A**, Wild-type germline showing DNA stained with DAPI (*top*), and immunostaining with phosphospecific HIM-3 antibodies (*middle*), and non-phosphospecific HIM-3 (pan-HIM-3) antibodies (*bottom*). **B**, Wild-type N2 oocyte precursor cells taken at different meiotic stages and immunostained for phospho-HIM-3 (green in merged images), SYP-1 and pan-HIM-3 (magenta in merged images). Diakinesis chromosomes (-1 diakinesis nucleus) with cruciform HIM-3 staining are circled in white. Due to the arbitrary orientation of chromosomes relative to the optical axis, cruciform structures of bivalent chromosomes are not always visible. **C**, Late pachytene cells (*gfp::cosa-1*) co-immunostained for phospho-HIM-3 (magenta in merged image), phospho-SYP-1 (green in merged image), and anti-GFP showing the position of GFP::COSA-1 (green in two middle images). Scale bars, 5μm.

of Ser277phos become confined to short arms in late pachytene, although some Ser277phos persists faintly on long arms until -2 diakinesis in the wild type. As described later in detail, *him-3(S282A)* mutants manifest a defect in establishing short/long arms, which hinders us from speculating on the localization of Ser277phos signals in the wild-type situation. These data suggest that differential regulation likely acts on Ser277 and Ser282 phosphorylation and dephosphorylation: Ser277 is more robustly phosphorylated than Ser282 from leptotene through mid pachytene, while Ser277phos confinement is less complete than that of Ser282phos.

## The density of phosphorylated HIM-3 increases on short arms upon partitioning

In addition to restriction of localization, HIM-3phos staining in wild type gonads appeared to increase in intensity once partitioning was observed (**Fig 3A**). To quantitatively measure this, we compared the levels of phosphorylated HIM-3 on short arms after partitioning with that of the entire chromosome axis before partitioning within the same gonad. Dividing late pachytene nuclei into pre- and post-partitioned regions, we picked 500 points (50 points in ten nuclei) on the SC in each region and recorded the intensity value per pixel after normalizing to the average intensity values of the pre-partitioning region. Our quantification showed that HIM-3phos signal intensity per pixel is higher on short arms after partitioning than along the entire axis before partitioning, indicating a net increase of phosphorylated HIM-3 on short arms (approximately 1.5 times increase, n = 6 gonads) (**Fig 3B, S4 Fig**). In addition, we compared panHIM-3 staining levels before and after HIM-3phos partitioning as well as short versus long arms after partitioning. We also found a slight (approximately 1.2 times increase, n = 6 gonads) but significant increase of panHIM-3 intensity on short arms after partitioning, suggesting a net addition of HIM-3 proteins to the SC toward the end of pachytene (**Fig 3B, S4 Fig**). This is consistent with the previous observation that HIM-3 and HTP-1/2 continue to accumulate on the SC throughout pachytene and diplotene [36]. In nuclei with partitioned HIM-3phos, panHIM-3 staining levels did not differ between short and long arms, indicating that the chromosome-wide increase in panHIM-3 levels does not derive from preferential addition of HIM-3 proteins to short arms but rather global addition along the entire SC (**Fig 3C, S4 Fig**). We reasoned that these observations could be interpreted in one of the following ways: (1) global dephosphorylation of HIM-3 with counter-acting phosphorylation specifically on short arms or (2) long arm-specific dephosphorylation and short arm-specific phosphorylation of HIM-3 (**Fig 3D**). Although other scenarios such as removal of phosphorylated HIM-3 proteins specifically from long arms with additional phosphorylation on short arms, or dephosphorylation from long arms and addition of phosphorylated HIM-3 on short arms are conceivable, we disfavor these possibilities since we did not detect any change in pan-HIM-3 staining levels between short and long arms after partitioning. However, we cannot exclude the possibility that the fraction of phosphorylated HIM-3 may be relatively small compared to the entire pool of axis-bound HIM-3, making the removal or addition of phosphorylated

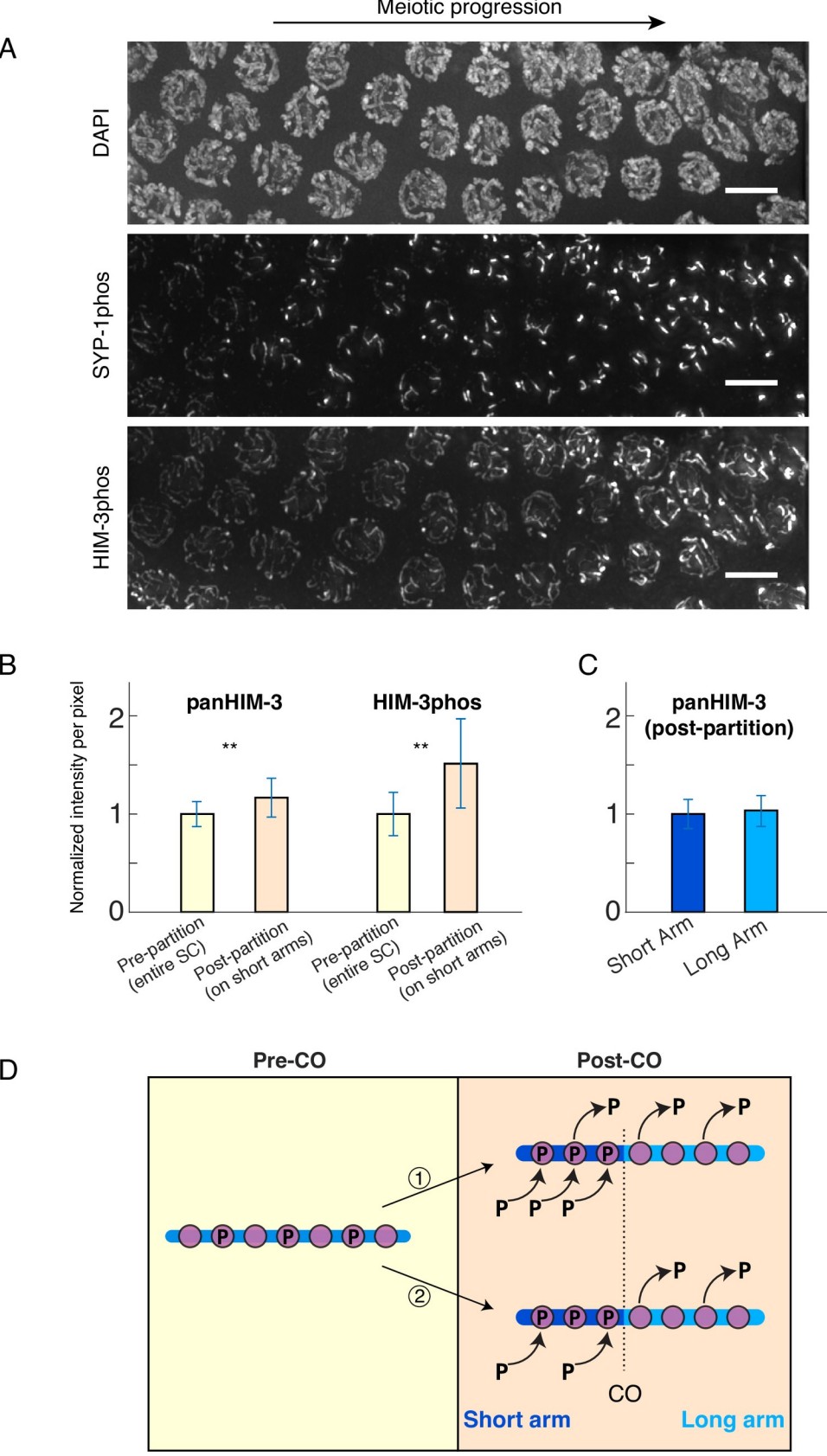

**Fig 3. HIM-3 phosphorylation increases on the short arm, while pan-HIM-3 increases slightly along the whole chromosome. A**, A segment of a wild-type germline (mid to late pachytene) showing meiotic progression from left to right. The accumulation of SYP-1 phosphorylation to one region (the short arm) can be seen to precede and exceed the partitioning of HIM-3. Scale bars, 5μm. **B**, Quantitation of pan-HIM-3 and phospho-HIM-3 staining in pre-partitioned chromosomes (yellow bars), post-partitioned chromosomes (beige, right). For both epitopes, values for post-partitioned chromosomes are significantly greater than that for pre-partitioned chromosomes (Shapiro-Wilk normality test followed by two-tailed T test, p<0.001). 3000 points for each condition (pre and post) are compared (50 points in each of 10 nuclei from 6 gonads); mean values per-nucleus for each set are shown in **S4 Fig**. **C**, Normalized intensity values of pan-HIM-3 in late pachytene nuclei comparing the short with the long arm. No significant difference exists between short and long arm staining (Shapiro-Wilk normality test followed by two-tailed T test). 3000 points for each condition (short and long) are compared (50 points in each of 10 nuclei from 6 gonads). Short arm points are the same points used for "post-partitioning" in **B**. For both **B** and **C**, intensity values derive from dividing raw pixel intensities of all individual pre- and post-partitioned points by the mean intensity of pre-partitioned points in the same gonad. Bars show mean values and error bars show standard deviation. **D**, Illustration of the two scenarios for phospho-HIM-3 enrichment discussed in the text. The blue line is the chromosome; violet circles stand for HIM-3 proteins; "P" indicates phosphorylation; arrows indicate the adding or leaving of molecules. On the right side, the short arm (dark blue) is toward the left of the vertical line indicating the crossover position (marked with "CO"); the long arm (light blue) is toward the right.

HIM-3 proteins undetectable by pan-HIM-3 staining. Since a previous photoconversion experiment showed that HIM-3 is rather static and stable once incorporated to the chromosome axis [37], we judge it unlikely that phosphorylated HIM-3 moves from long to short arms. Taken together, our data suggest that short arm enrichment of phosphorylated HIM-3 is likely to occur through a combination of dephosphorylation of phosphorylated HIM-3 from long arms and short arm-specific phosphorylation activity.

Since the activity of PP1 phosphatase, GSP-2, has been shown to localize to long arms upon crossover designation via HTP-1/2 and LAB-1 interactions [23,24], we next tested whether PP1 phosphatase dephosphorylates HIM-3 on long arms by immunofluorescence. First, in *gsp-2(tm301)* mutants, HIM-3-phos was still enriched on short arms and absent from long arms, showing no difference from wild-type staining. Furthermore, simultaneous degradation of GSP-2 and its paralog GSP-1 by the auxin-induced degron (AID) system [38] also showed timely enrichment of HIM-3phos to short arms. These data suggest that GSP-1/2 phosphatases are not responsible for loss of phosphorylated HIM-3 from long arms (**S4 Fig**).

To understand the mechanism of the putative kinase activity on short arms, we examined the localization of phosphorylated HIM-3 in various meiotic mutant backgrounds. Since Ser282 is in an ATM/ATR kinase consensus site (SQ), we examined if Ser282 is phosphorylated by these kinases in the *him-3(S277A)* mutant background. We found that phosphorylation of HIM-3 Ser282 does not depend on these DNA damage-responsive kinases (**S4 Fig**).

Unexpectedly, we found that HIM-3 phosphorylation is dependent on synapsis (**S5 Fig**). In *syp-1(me17)* null mutants or other meiotic mutants such as *plk-2(ok1936)* that generate unsynapsed chromosomes [39–41], HIM-3phos staining was completely absent from unsynapsed chromosomes. This suggests that recruitment or activation of one or more kinases that phosphorylate HIM-3 is dependent on SYP proteins. Since a Polo-like kinase, PLK-2, is known to be recruited to the SC by phosphorylated SYP-1 [20], and PLK-1 and PLK-2 redundantly phosphorylate SYP-4 on the SC [42], we examined if PLK-1 and PLK-2 also phosphorylate HIM-3 by immunofluorescence. In *plk-1(RNAi); plk-2(ok1936)* mutant gonads, homologous synapsis was significantly delayed, but HIM-3phos antibody signals were still detected on synapsed chromosomes (**S5 Fig**). Effectiveness of *plk-1* RNAi was verified by embryonic inviability and mitotic defects both in the mitotic region of the gonad as well as in embryos. We also investigated the kinase CHK-2, a key meiotic kinase involved in chromosome pairing, synapsis, double strand break formation and meiotic checkpoint with several known targets in early meiotic prophase. We found that HIM-3 is phosphorylated normally on synapsed

chromosomes in *chk-2(me64)* mutants (**S5 Fig**). Further investigation is needed to identify the Ser/Thr kinase(s) as well as the Tyr kinase, which phosphorylate HIM-3 at Ser277, Tyr279 and Ser282.

## Asymmetric enrichment of phosphorylated HIM-3 is dependent on crossover designation

Next, we tested if asymmetric enrichment of phosphorylated HIM-3 is dependent on the formation of crossover intermediates. DNA double strand breaks (DSBs) are created by SPO-11, a topoisomerase-like enzyme, to initiate homologous recombination. In *spo-11(me44)* mutants, no programmed DSBs are generated during meiotic prophase and thus meiocytes fail to form crossovers [43]. In *spo-11 (me44)* mutants, the great majority of nuclei lack both DSBs and crossover designation marker COSA-1, and show HIM-3phos staining along the entire length of the SC even at very late pachytene (**Fig 4A**). To further test whether crossover intermediates *per se* and not only DSBs are required for enrichment of phosphorylated HIM-3, we visualized phospho-HIM-3 and SYP-2 in *msh-5* mutants, which achieve even higher DSB levels than wild-type, but do not form crossover intermediates [44,45]. We found no partitioning of phospho-HIM-3 in *msh-5* mutants, indicating that crossover intermediates containing MSH-5 are required for the enrichment (**Fig 4B**).

Previous studies showed that a small minority of chromosomes in *spo-11 (me44)* mutants contain COSA-1 foci, which presumably result from unrepaired DNA damage incurred during replication; however, these foci are not able to mature into genuine crossovers [46,47]. The same studies also have shown that chromosomes with such COSA-1 foci become strongly enriched for SYP-1 and its interacting protein PLK-2 in a chromosome-autonomous manner, at the expense of chromosomes that lack COSA-1 foci [46,47]. Similarly, we found that rare chromosomes with COSA-1 foci in *spo-11(me44)* mutants show enrichment of phosphorylated HIM-3 along the entire length of the chromosome (**Fig 4,** the larger gonad area is also shown in **S6 Fig**). In these same nuclei, HIM-3phos staining was mostly absent from chromosomes lacking COSA-1. However, pan-HIM-3 staining intensity appeared the same between chromosomes with and without COSA-1 foci. Taken together, these observations suggest that HIM-3 is phosphorylated on chromosomes bearing COSA-1 foci, and dephosphorylated on chromosomes without COSA-1.

Next, we examined whether phosphorylation of other SC components displayed a similar pattern to HIM-3phos in nuclei with a single COSA-1 focus. We found that phosphorylated SYP-1 at Thr452 also becomes enriched along the entire length of chromosomes harboring COSA-1 foci in *spo-11(me44)* before the bulk of SYP-1, which was visualized by pan-SYP-1 antibody (**Fig 4C**). This is consistent with a previous observation that PLK-2 accumulates on chromosomes with DNA breaks, and with our previous evidence suggesting that phosphorylated SYP-1 at Thr452 recruits PLK-2 to the SC [20,47]. These results add to the growing evidence that the presence of recombination intermediates containing MSH-5 (i.e., crossover intermediates in wild-type animals) alters the chromosome environment in *cis*, and enrichment of phosphorylated HIM-3 and SYP-1, compared to their unphosphorylated forms, is more responsive to this chromosome-wide change.

## Asymmetric enrichment of phosphorylated HIM-3 is sensitive to the number of crossovers

The observed enrichment of phosphorylated HIM-3 and SYP-1 in rare *spo-11* nuclei that contain a single chromosome with a COSA-1 focus differs from the wild-type situation in one important respect: both phosphoproteins fail to partition to the short arm of the chromosome,

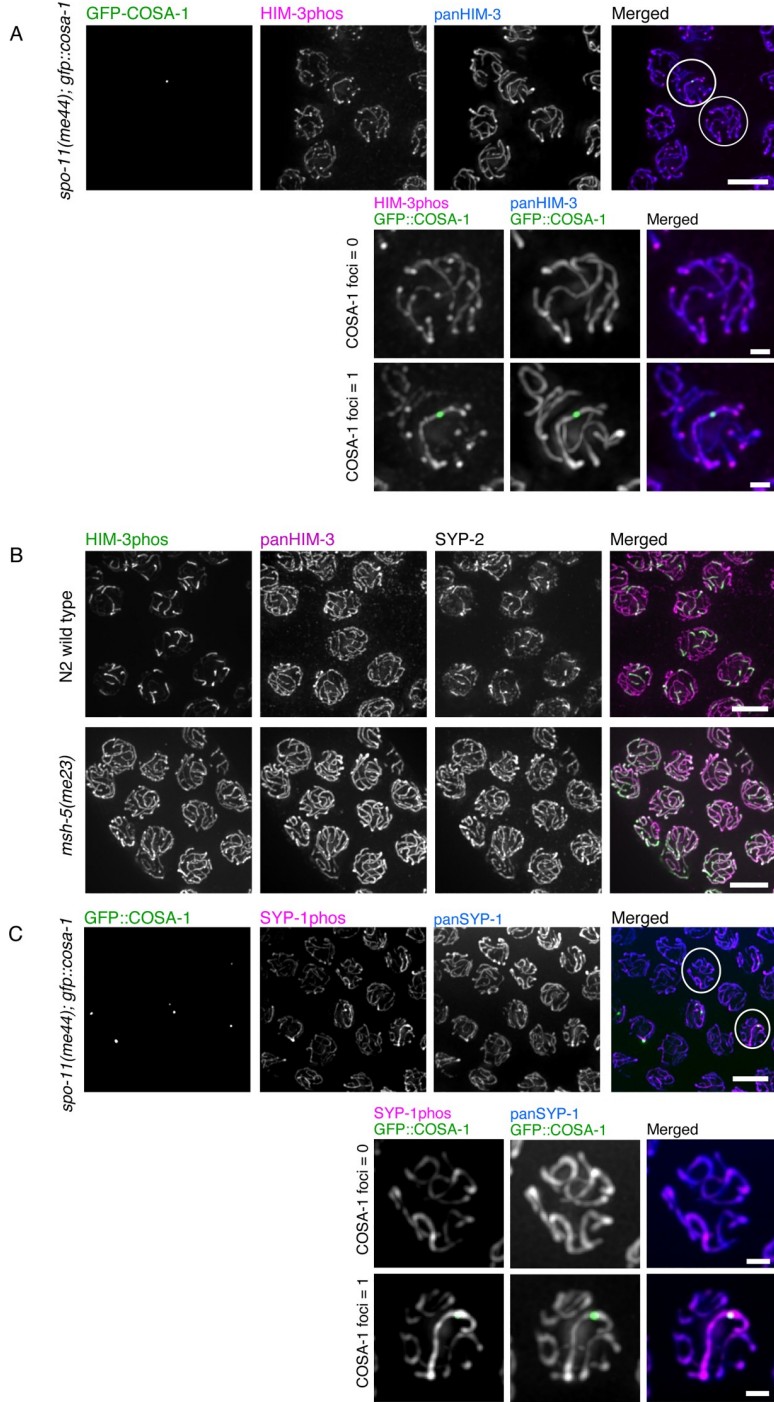

**Fig 4. Asymmetric localization of phosphorylated HIM-3 requires crossover intermediates. A**, Immunostaining of crossover designation marker GFP-COSA1 (green in merged image), phospho-HIM-3 (magenta in merged image), and pan-HIM-3 (blue in merged image) in *spo-11(me44); gfp::cosa-1* oocyte precursors. **B**, Immunostaining of GFP-COSA1 (green in merged image), phospho-SYP-1 (magenta in merged image), and pan-SYP-1 (blue in merged image) in *spo-11(me44); gfp::cosa-1* oocyte precursors. For both **A** and **B**, at top right, white circles surround nuclei shown magnified in insets below; COSA-1 focus number (0 or 1) is indicated. **C**, Wild-type (top row) and *msh-5* (bottom row) germlines at late pachytene stained with antibodies against phospho-HIM-3 (green in merged images), pan-HIM-3 (magenta in merged images), and SYP-2 (gray; not shown in merged images). Scale bars: 5μm (unmagnified), 1μm (magnified).

but rather remain present along the entire chromosome length. We reasoned this difference could be due either to the differing physical nature of DSBs induced by unrepaired mitotic damage in *spo-11* mutants, compared to SPO-11-catalyzed DSBs, or to the lower number of crossover intermediates present. To distinguish these possibilities, we examined *dsb-2* mutants, in which DSBs are still catalyzed by SPO-11 but are present at greatly reduced levels, giving rise to meiocytes with varying numbers of DSBs and thus crossovers (ranging from 0 to 6 COSA-1 foci) in the same gonad [48]. In *dsb-2* mutants, we observed partitioning of HIM-3phos to short arms, but only in nuclei with relatively high numbers of crossover intermediates. Complete partitioning of HIM-3phos was detected only in nuclei with 5 or 6 COSA-1 foci in late pachytene, whereas accumulation along the entire length of chromosomes was seen in nuclei with 4 or fewer COSA-1 foci. The fraction of nuclei with partial (some but not all chromosomes with a COSA-1 focus achieved partitioning) or complete partitioning of HIM-3phos increased as meiotic prophase progressed, and in diplotene, about 20% of nuclei with 4 COSA-1 foci also achieved complete partitioning (**Fig 5A; quantification of 5A is presented in 5C left panel**). In contrast, complete partitioning of SYP-1phos was already achieved in late pachytene in many nuclei with 4 or more COSA-1 foci (44% of nuclei with CO = 4, and 73% of nuclei with CO = 5), and no partitioning was observed in nuclei with 1, 2 or 3 COSA-1 foci (**Fig 5B and 5C right panel**). Since SYP-1 disassembles completely from long arms in diplotene, our quantification of SYP-1phos partitioning was limited to late pachytene nuclei but did not include diplotene. While we frequently observed a high proportion of partial partitioning of HIM-3phos, partial partitioning of SYP-1phos was relatively rare, suggesting that partitioning of SYP-1phos is a more rapid, switch-like process involving feedback mechanisms. These results show that the overall number of crossover intermediates in a nucleus can cumulatively influence the efficiency of short-versus-long arm partitioning.

## Phosphoregulation of HIM-3 promotes timely SC disassembly

To understand the function of HIM-3 phosphorylation, we used CRISPR to generate non-phosphorylatable as well as phospho-mimetic *him-3* mutants at the original locus. For non-phosphorylatable mutants, Ser277, Tyr279 and Ser282 were respectively mutated to Ala, Phe and Ala either solely (S282A mutant) or in combination (FA mutant: Y279F+S282A; AFA mutant: S277A+Y279F+S282A). HIM-3 peptides carrying the S282A point mutation retained significant HTP-2 binding capacity in the fluorescence polarization binding assay, although with a slightly lowered binding constant (**Fig 1C:** HTP-2 binding Kd values were: 5.8μM for unphosphorylated peptides v.s. 11.3μM for S282A peptides; two-tailed P value<0.0001 by unpaired t-test). Slightly reduced HTP-1 levels were detected in *him-3(S282A)* diakinesis nuclei *in vivo* (**S8 Fig**). To create phosphomimetic mutants, Ser282 was mutated either to Glu or Asp (S282E or S282D mutant). Some of the point mutations were generated in a strain containing a C-terminally FLAG-tagged *htp-1* (*htp-1::flag)* at the endogenous locus to facilitate HTP-1 visualization by immunofluorescence. The S282D mutation on its own, as well as the S282E mutation in an *htp-1::flag* background, showed modest, but statistically significant levels of embryonic inviability and male production, indicating a problem with meiosis. However, the addition of a FLAG tag did not confer any meiotic defects, as this strain showed normal embryonic viability (**S6 Fig**) and did not enhance meiotic phenotypes. In all of the *him-3* mutants, HIM-3 proteins are expressed at normal levels by western blot (**S6 Fig**) and localize to the SC normally by pan-HIM-3 antibody staining (**S7 Fig**). In all *him-3* phosphomutants, HTP-1 was detected on the SC from the leptotene/zygotene transition zone, and SC central elements were polymerized normally (**S7 Fig**). However, as expected for phosphomimetic mutants, quantitative image analysis showed that global HTP-1::flag levels in *him-3(S282E)*

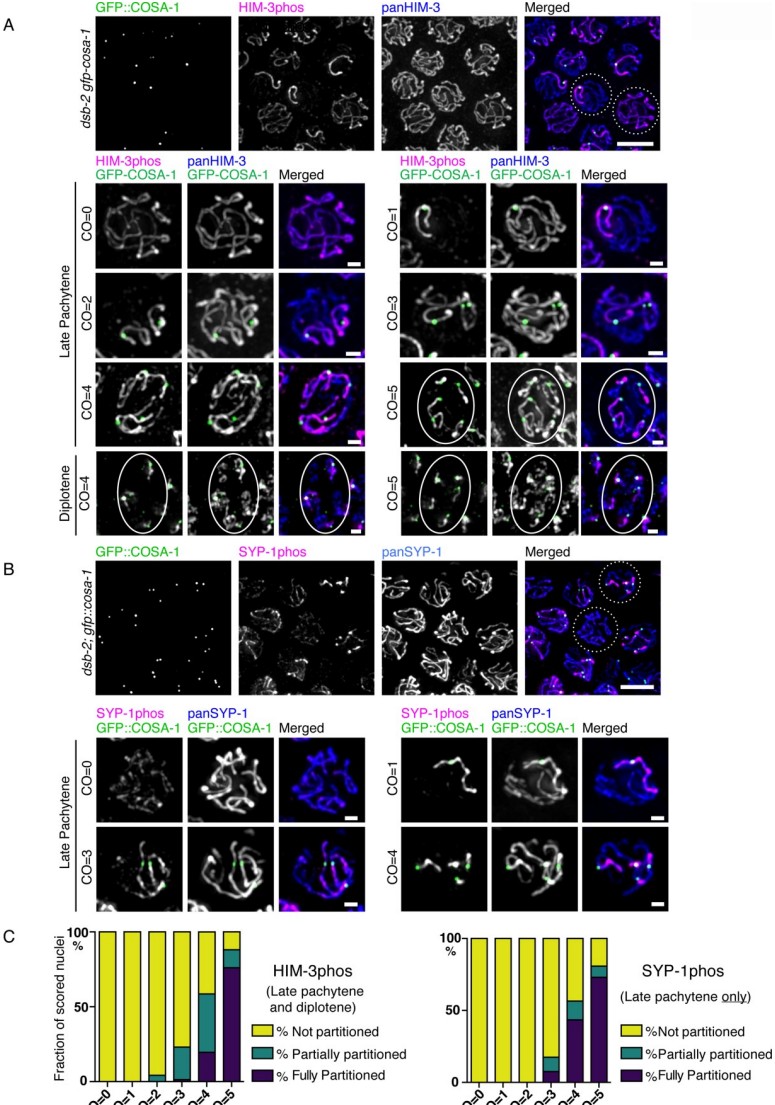

**Fig 5. Asymmetric localization of phosphorylated HIM-3 depends on the number of crossover intermediates. A**, *Top*: Immunostaining of crossover designation marker GFP-COSA1 (green in merged image), phospho-HIM-3 (magenta in merged image), and pan-HIM-3 (blue in merged image) in *dsb-2(me96); gfp::cosa-1* oocyte precursors (late pachytene). Dotted-circled nuclei are shown magnified in the top row below. *Bottom*, individual nuclei with different numbers of COSA-1 foci (left labels). Ellipses surround those nuclei with asymmetrically partitioned phospho-HIM-3. **B**, *Top*: Immunostaining of GFP-COSA1 (green in merged image), phospho-SYP-1 (magenta in merged image), and pan-SYP-1 (blue in merged image) in *dsb-2(me96); gfp::cosa-1* oocyte precursors (late pachytene). *Bottom*, individual nuclei with different numbers of COSA-1 foci (left labels). Scale bars: 5μm (overview images), 1μm (individual nuclei). **C**, Quantitation of observed nuclei stained with either phospho-HIM-3 (left, n = 243) or phospho-SYP-1 (right, n = 164) showing classification of partitioning.

*htp-1*::*flag* were severely reduced in diakinesis compared to control gonads (*htp-1*::*flag* by itself) dissected and immunostained side-by-side on the same slide (**S8 Fig**). This confirms that *in vivo*, some fraction of HTP-1 does indeed normally bind to HIM-3, presumably the non-phosphorylated pool.

Since the fluorescence polarization assay suggested that phosphorylation of HIM-3 at S282 prevents HTP-1/2 binding, and phosphorylated HIM-3 is enriched on short arms from which

HTP-1/2 is removed, we wondered whether HIM-3 phosphorylation prevents HTP-1/2 from re-binding to short arms once they dissociate from the SC. To test this, we examined the confinement of HTP-1 to long arms as well as confinement of SYP-1 to short arms in diakinesis nuclei in *him-3* phospho-mutants. Since condensed chromosomes at diakinesis can be difficult to resolve with light microscopy due to the arbitrary orientation of chromosome arms relative to the optical axis, we limited our analysis to bivalent chromosomes in which we could clearly resolve a cruciform structure of HTP-3, which localizes to both short and long arms. Within each nucleus, we scored as many resolvable bivalent chromosomes as possible, and counted the nucleus as positive if at least one bivalent showed abnormal localization of SYP-1 or HTP-1/2. Since SYP-1 dissociates from chromosomes and cannot be detected reliably in -3, -2 and -1 diakinesis nuclei in wild type, the analysis was limited to -7 through -4 diakinesis nuclei. If HIM-3 phosphorylation normally functions to prevent HTP-1/2 from re-binding to short arms, non-phosphorylatable *him-3* mutants would be expected to retain HTP-1/2 on short arms. However, we did not detect any delay in HTP-1 dissociation from short arms in *him-3 (S282A)* or *(AFA)* mutants (**Fig 6A–6C,** only quantification is shown for S282A). To our surprise, instead, phosphomimetic mutants (S282D or S282E) delayed HTP-1 dissociation from short arms on at least one chromosome in about 25% of oocytes, which show persistent HTP-1 on short arms in early diakinesis, while HTP-1 became completely restricted to long arms during the earlier diplotene stage in control gonads (**Fig 6**). Although HTP-1 persisted longer on short arms on some bivalents in *him-3* phosphomimetic mutants until -3 diakinesis, eventually HTP-1 was confined to long arms on all the bivalents in -1 diakinesis nuclei (**Fig 6D, S9 Fig**), suggesting that partitioning of phosphorylated HIM-3 influences the timing of HTP-1 dissociation from short arms but is not absolutely required. Unexpectedly, both non-phosphorylatable and phosphomimetic *him-3* mutants also delayed SYP-1 confinement to short arms and showed persistent SYP-1 on long arms in the majority of nuclei through diakinesis, while in the wild type SYP-1 is completely confined to short arms in early diplotene. Although SYP-1 persisted on both arms in many *him-3* mutant nuclei, SYP-1 eventually dissociated from chromosomes in late diakinesis nuclei with normal timing (-2 or -3 diakinesis and onward)(**Fig 6D, S9 Fig**). We verified that the delayed establishment of SYP-1/HTP-1 asymmetry in our *him-3* phospho mutants is not due to slower cell cycle progression at the transition from diplotene to diakinesis by immunofluorescence against OMA-2, a translational regulator expressed in maturing diakinesis oocytes [49] (**S10 Fig**). Our results suggest that partitioning of phosphorylated HIM-3 to short arms promotes global rearrangement of SC components, including central elements, along the entire length of the SC rather than locally on short arms. Further, the normal dissociation of HTP-1 from the short arm in *him-3* non-phosphorylatable mutants shows that HIM-3 phosphorylation is not required to prevent HTP-1/2 from re-binding the short arm in diplotene after their dissociation.

Since asymmetric SC disassembly is a precursor to establishment of chromosome separation domains, and previous studies on mutations that perturb asymmetric SC disassembly also mislocalized CPC activity [3,15,20,23,32], we examined CPC localization on short arms by immunofluorescence of CPC component ICP-1 (the *C. elegans* INCENP ortholog) in *him-3* mutants. In wild type, ICP-1 is detected in -3, -2, -1 diakinesis nuclei and later, is strongly enriched on short arms in -3 and -2 diakinesis, and becomes fully confined to short arms from -1 diakinesis through metaphase I oocyte pronuclei (**S11 Fig**). In *him-3(S282D)* mutants, ICP-1 short arm confinement was reduced, but S282A, AFA or S282E mutants did not show a significant difference from control animals. Also, consistent with ICP-1 localization, embryonic viability and the percentage of male progeny (indicative of *X* chromosome meiotic nondisjunction) are very similar to wild type in non-phosphorylatable mutants, and only a slight increase in embryonic lethality and male progeny number are observed in phospho-mimetic

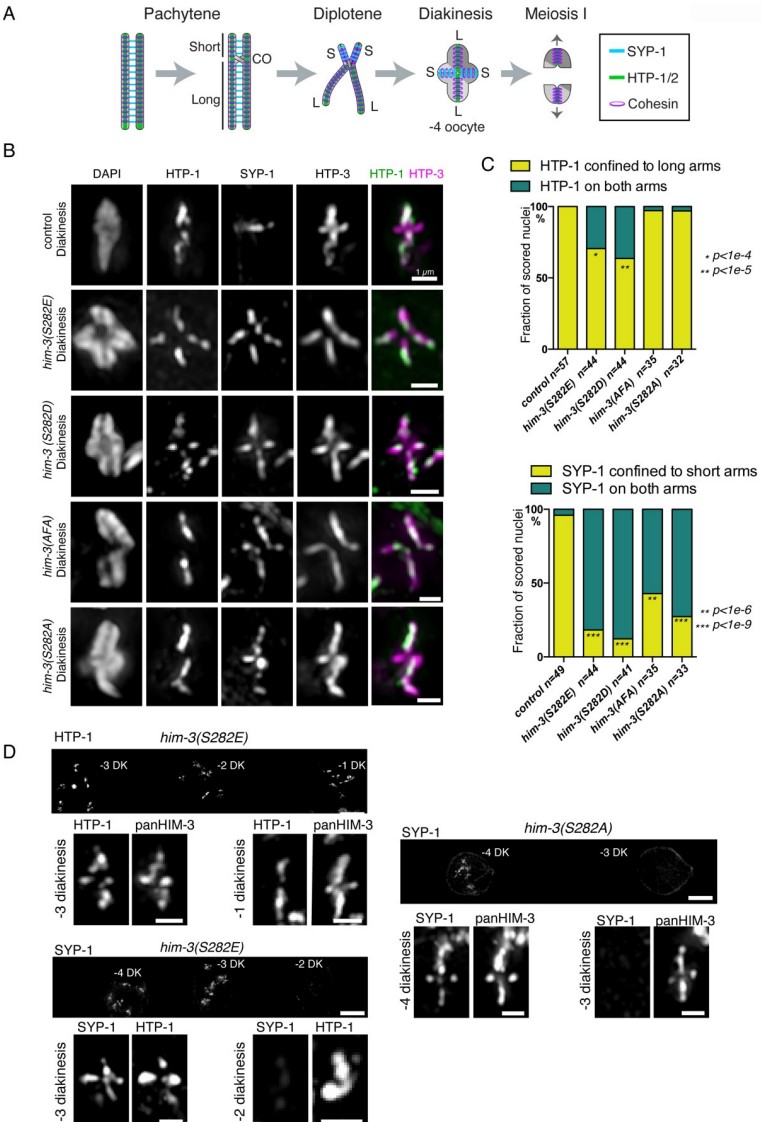

**Fig 6. Phosphoregulation of HIM-3 is involved in timely SC disassembly. A**, Diagram of chromosome remodeling after crossover designation, showing normal localization of SYP-1 and HTP-1/2 at diakinesis. **B**, Immunostaining of HTP-1 (green in merged images), SYP-1, and HTP-3 (magenta in merged images) on -4 to -7 diakinesis nuclei of *htp-1::flag*, *him-3(S282E) htp-1::flag*, *him-3(S282D)*, *him-3(AFA) htp-1::flag*, and *him-3(S282A) htp-1::flag*. Single chromosome images were cropped out of 3D data sets and shown as partial projections. Scale bars, 1µm. **C**, Quantitation of partitioning degree of HTP-1 (top) and SYP-1 (bottom) for *htp-1::flag*, *him-3(S282E) htp-1::flag*, *him-3(S282D)*, *him-3(AFA) htp-1::flag* and *him-3(S282A) htp-1::flag*. The number of diakinesis nuclei scored is shown beneath the corresponding bar. Indicated p values are from a chi-squared test for independence from the control values, with Bonferroni correction. **D**, Immunostaining of HTP-1 and SYP-1 in -1 through -4 diakinesis nuclei of *him-3(S282E) htp-1::flag* or *him-3(S282A) htp-1::flag*. In *him-3(S282E) htp-1::flag* mutants, HTP-1 and SYP-1 persist on both arms on some bivalents until -3 or -4 diakinesis, but eventually HTP-1 becomes confined to long arms while SYP-1 dissociates from chromosomes by -2 or -1 diakinesis. Similarly in *him-3(S282A) htp-1::flag* mutants, SYP-1 persists on both arms on some bivalents until -3 or -4 diakinesis but dissociates from chromosomes by -2 or -1 diakinesis. Scale bars, 5µm, or 1µm for the magnified bivalent panels. Partially projected images are shown for the magnified bivalents. The same images with counterstaining of DAPI and panHIM-3 to show chromosome axis, and their control images are also provided in **S9 Fig**.

mutants (0.8% embryonic lethality in control v.s. 5.1% lethality in S282E, 0.3% lethality in control v.s. 2.2% lethality in S282D: **S6 Fig**). This suggests that although SYP-1 dissociation is

significantly delayed in all *him-3* phospho-mutants, and asymmetric distribution of HTP-1 is delayed in *him-3* phospho-mimetic mutants, this does not lead to gross perturbation in designating chromosome separation sites, perhaps because other pathways including ZHP-1/2, phosphorylated SYP-1 and PLK-2 [15], akirin [50], or MAP kinase and phosphorylated SYP-2 [19] redundantly function to designate the separation site.

While we characterized the *him-3(282A)* mutant, we noted that Ser277phos staining visualized by HIM-3phos antibody persisted evenly and robustly on both short and long arms after CO designation throughout meiotic prophase, including -1 diakinesis (**S3 Fig**). This is a striking contrast to HIM-3phos antibody staining in wild type, in which the bulk of HIM-3phos signals are enriched on short arms and only very faint HIM-3phos signals are detectable on long arms until -2 diakinesis (**S1 Fig**). This suggests that the S282A mutation not only delayed asymmetric disassembly of SYP-1 and HTP-1 but also abrogated Ser277 dephosphorylation on long arms. These data together reinforce the idea that phosphorylation at HIM-3 Ser282 promotes global rearrangement of SC components, including phosphorylated S277 on HIM-3 itself.

## Partitioning of phosphorylated HIM-3 depends on SYP-1 Thr452 phosphorylation

Since the timing of phosphorylated HIM-3 enrichment on short arms follows confinement of phosphorylated SYP-1, we next asked whether HIM-3phos enrichment is a downstream event of SYP-1phos confinement or a parallel pathway for asymmetric SC disassembly. In non-phosphorylatable *syp-1(T452A)* mutants or *plk-2(ok1936)* mutants, HIM-3 was phosphorylated normally on synapsed chromosome regions beginning at the leptotene/zygotene transition zone, but phosphorylated HIM-3 failed to accumulate on short arms in late pachytene (**S12 Fig**). This suggests that while initial phosphorylation of HIM-3 does not require SYP-1 phosphorylation at Thr452 and subsequent PLK-2 loading, partitioning of phosphorylated HIM-3 to the short arm does require SYP-1 phosphorylation. This is consistent with our prior observation that short arm enrichment of HIM-3phos happens later than SYP-1phos confinement but earlier than bulk SYP-1 confinement (**Figs 2B and 3A, S1 Fig**). In *plk-2 (ok1936)* mutants, SYP proteins do not disassemble asymmetrically but remain on both arms until the end of pachytene and abruptly accumulate at CO sites in diplotene and diakinesis [40]. Similarly to SYP proteins, in *plk-2(ok1936)* mutants, phosphorylated HIM-3 remained evenly distributed along the entire SC until the end of late pachytene, and from diplotene became colocalized with SYP-2 at foci representing presumptive CO sites (**S12 Fig**).

In contrast, in *him-3(S282E)* or (*AFA*) mutants, phosphorylated SYP-1 was partitioned normally to short arms (**S13 Fig**). These observations suggest that partitioning of phosphorylated HIM-3 is a downstream event of phosphorylated SYP-1 partitioning, and in turn, promotes dissociation of SYP-1 from long arms. This observation shows that partitioning of SYP-1 phosphorylation to the short arm is not sufficient to remove SC central element proteins from the long arm. Consistent with the idea that SYP-1 phosphorylation is epistatic to HIM-3 phosphorylation with regard to establishing short/long arm asymmetry, double mutants of *syp-1* non-phosphorylatable mutation and *him-3* phospho-mutation (*syp-1(T452A; him-3(S282A)*, *syp-1(T452A); him-3(Y279A S282A)*, *syp-1(T452A); him-3(S282E)*) showed similar embryonic viability with *syp-1(T452A)* single mutants (**S13 Fig**).

Somewhat unexpectedly, we noted that in -4 and -3 diakinesis nuclei, weak phosphorylation of SYP-1 was observed on long arms in *him-3* phospho-mutants, although it was strictly limited to short arms at earlier time points (**S13 Fig**). Since SYP-1 normally begins to dissociate from the chromosome axis at -4 diakinesis, it is unlikely that phosphorylated SYP-1 proteins

are newly added to long arms in these mutant nuclei. Rather, hyperpersistent SYP-1 on long arms in *him-3* phospho-mutants is likely phosphorylated *de novo* by a kinase that is activated only in mature diakinesis nuclei. In the wild type, since SYP-1 is already limited to short arms by diakinesis, this kinase activity would not lead to accumulation of phosphorylated SYP-1 on long arms.

Altogether, our results suggest that HIM-3phos partitioning is a downstream event of SYP-1phos partitioning. Since disruption of SYP-1 phosphorylation leads to stronger perturbations in HTP-1/2 localization, CPC localization and the first meiotic division [20] than seen in *him-3* phospho-mutants, we conclude HIM-3phos partitioning is one of many downstream events depending on SYP-1phos partitioning.

## Phosphoregulation of HIM-3 is not required to prevent recombination between sister chromatids nor to satisfy the synapsis checkpoint

During meiotic prophase, programmed DSBs are preferentially repaired via inter-homolog recombination to generate crossovers between homologs while inter-sister recombination is actively suppressed by a mechanism depending on, among other proteins, HIM-3 and HTP-1/2 [29,51]. In various organisms, HORMA domain proteins are required for homolog-biased recombination, and specifically phosphorylation of the budding yeast HORMA domain protein, Hop1, by ATM/ATR kinases has been shown to be important for inter-homolog bias [52–56]. We therefore wondered whether HIM-3 phosphorylation functions to prevent inter-sister recombination during meiotic prophase. In *C. elegans*, sister chromatid-mediated homologous recombination requires BRC-1 (the *C. elegans* ortholog of BRCA1) and its inter-actor BRD-1 (BARD1 ortholog) [57–60], and mutants defective in preventing inter-sister recombination during meiotic prophase synergistically increase embryonic inviability when combined with *brc-1* or *brd-1* mutants [61]. To test whether *him-3* phosphorylation might block the use of the sister chromatid as a repair template, we combined our non-phosphorylatable *him-3* mutation with *brc-1* and *brd-1* mutations by generating triple mutants, *brc-1 (tm1145) brd-1(dw1); him-3 (S282A)* or *brc-1(tm1145) brd-1(dw1); him-3(AFA)*, and scored embryonic viability. However, combining these mutations did not result in synthetic embryonic lethality (**S14 Fig**). Similarly, *him-3(AFA)* and *(S282E)* mutations had no effect on progeny viability of 75 Gy γ-irradiated animals (**S14 Fig**), showing it is not defective in sister chromatid-based DNA repair caused by excess DNA damage [61,62].

The dependence of HIM-3 phosphorylation on synapsis suggested the possibility of its acting to satisfy the synapsis checkpoint, which subjects cells that fail synapsis to programmed cell death [63] and depends on chromosome axis proteins, including HIM-3 and HTP-1/2 [63–65]. However, we saw no significant increase in apoptotic nuclei in *him-3 (S282A)* or *(Y279F S282A)* germlines compared to controls, indicating that this phosphorylation is not required to satisfy the synapsis checkpoint (**S14 Fig**). Taken together, these results indicate that HIM-3 phosphorylation at the closure motif is not involved in either the regulation of inter-homolog bias for homologous recombination or in the synapsis checkpoint.

## Discussion

Here, we have identified and characterized phosphorylation sites within the closure motif of the conserved HORMA domain protein HIM-3. We showed that phosphorylated HIM-3 is enriched on short arms, and this partitioning contributes to the asymmetric disassembly of SYP-1 and HTP-1/2. Since *him-3* phospho-mutants show a delay in asymmetric SC disassembly, HIM-3 phosphorylation is not just a passive phosphorylation as a consequence of being unbound by HTP-1/2, but is likely a part of a redundant collection of mechanisms ensuring

establishment of short and long arms prior to the stepwise cohesin degradation at meiotic divisions (phenotypes summarized in **S15 Fig**).

By preventing association of HTP-1/2 with HIM-3, phosphorylation of the HIM-3 closure motif may functionally modulate the axis structure. A previous study has shown that HTP-1/2 is able to bind either to HIM-3 or HTP-3's closure motifs [27]. A more recent study has shown that on average two HTP-1/2 molecules and three HIM-3 molecules are present at every HTP-3 molecule, with substantial variation between HTP-3 molecules [36]. This indicates that closure motifs are rarely if ever occupied to their theoretical maximum; instead, many must remain unbound. The diverse, multivalent assembly that this scheme allows may be important to provide flexibility for regulating SC function.

In yeast and mice, SC disassembly upon pachytene exit is triggered by Polo kinase (Cdc5), DDK, and Aurora B kinase [66–69]. It is predicted that phosphorylation of SC components by these kinases leads to destruction of the SC by proteolysis, but the responsible phosphorylation sites on SC components and molecular mechanisms of SC disassembly have not been identified yet. A previous study has shown that disruption of binding between a human HORMAD protein, Mad2, and its binding partner CDC20 containing the closure motif requires unfolding of the N-terminal region of the HORMA domain by the AAA+ ATPase TRIP13, and thus requires energy consumption [70]. Currently, details of the molecular mechanism promoting dissociation of HTP-1/2 from HIM-3 or HTP-3 on short arms, or whether dissociation requires consumption of energy or proteolysis, are lacking. Since our *him-3* non-phosphorylatable mutants showed normal dissociation of HTP-1/2 from short arms, it is unlikely that HIM-3 phosphorylation at the closure motif functions to prevent re-binding of dissociated HTP-1/2 upon crossover designation. Rather, phosphomimetic mutations in HIM-3 delayed HTP-1 disassembly from short arms. If, as we speculate, HIM-3 phosphorylation prevents HTP-1/2 binding *in vivo*, HTP-1/2 will likely be incorporated to the axis solely by binding to HTP-3 closure motifs 1 and/or 6 in *him-3* phospho-mimetic mutants. In this scenario, we speculate that dissociation of HTP-1/2 may be less efficient when thus "overloaded" onto HTP-3 compared to when bound to both HTP-3 and HIM-3, implying that HIM-3 may provide a binding site for HTP-1/2 to be readily disassembled upon crossover designation in the wild type. A previous study has shown that disrupting the two HTP-1/2 binding sites of HTP-3 causes HTP-1/2 to bind exclusively to HIM-3 and leads to a delay in synapsis [27], suggesting the possibility that a fraction of HIM-3 protein must be unoccupied by HTP-1/2 for timely polymerization of SYP-1. HIM-3 phosphorylation at its closure motif may prevent saturation of HIM-3 by HTP-1/2 and thus may promote timely interactions between central and axial elements of the SC. Although we cannot exclude the possibility that our phosphomimetic mutations may not completely mimic phosphorylation and may produce a neomorphic phenotype of delayed SC disassembly, the asymmetric localization of phosphorylated HIM-3 on short arms in wild type gonads suggests that this phosphoregulation is likely involved in asymmetric disassembly of the SC.

In addition to delayed HTP-1/2 disassembly, dissociation of unphosphorylated SYP-1 from the long arm is delayed in both *him-3* non-phosphorylatable and phospho-mimetic mutants. Although we lack direct evidence that central elements directly bind to HIM-3, previous studies suggest that HIM-3 is likely the interface between central elements and the chromosome axis: superresolution microscopy analysis has shown that among the chromosome axis proteins, HIM-3 localizes most closely to central elements [71]. Point mutations at the HIM-3 closure motif may affect interactions between HIM-3 and central elements, leading to prolonged synapsis in *him-3* phosphomimetic or non-phosphorylatable mutants. Alternatively, partitioning of phosphorylated and non-phosphorylated fractions of HIM-3 may normally promote asymmetric dissociation of SYP proteins from the SC.

Although SC disassembly is delayed in *him-3* phospho-mutants, this does not lead to gross perturbations of two-step cohesion loss and meiosis segregation, as only a slight reduction in embryonic viability was observed in *him-3* phosphomimetic mutants. This is consistent with previous results that a *him-3* C-terminal deletion lacking the entire closure motif does not show major meiotic defects [27]. This observation adds to the growing evidence that redundant mechanisms ensure CPC localization and limit cleavage of cohesin to short arms at meiosis I in the absence of asymmetric disassembly of the SC. We have previously shown that loss of SYP-1 Thr452 phosphorylation prevents asymmetric dissociation of HTP-1/2 and SYP-1, yet reduces embryonic viability only mildly (39.7% reduction in *syp-1(T452A)*). Similarly, *syp-2* phosphomimetic mutants, in which approximately 50% of bivalents have persistent SYP-1 on both short and long arms in diakinesis, have only a 14.3% reduction in embryonic viability [19]. In these mutants, slight enrichment of SYP-1 on short arms and HTP-1/2 on long arms relative to the other arm may be sufficient to limit the CPC activity to short arms, since CPC localization is known to be regulated through both positive and negative feedback [72]. Alternatively, other mechanisms independent of asymmetric SC disassembly may exist to ensure concentration of CPC activity to short arms at meiosis I. For example, since metaphase I bivalents orient with the long arms pointing toward opposite spindle poles, concentration of CPC components at the spindle midzone would automatically favor destruction of cohesin on the short arm. The specific action of each mechanism toward the final outcome of two-step chromosome disjunction remains to be determined.

Subthreshold levels of crossover intermediates are known to activate a surveillance system causing enrichment of SYP-1 in *cis* onto chromosomes with crossover intermediates along the entire length of the SC, while SYP-1 dissociates from chromosomes without crossover intermediates [46,47]. Our data have shown that this enrichment of phosphorylated SYP-1 precedes that of bulk SYP-1 in *spo-11* or *dsb-2* mutants, indicating that phosphorylated form is more responsive to SC alteration triggered by crossover designation. However, how this crossover intermediate-induced stabilization of SYP-1 relates to partitioning of SYP-1 on short arms at the molecular level is not understood. Our analysis of *dsb-2* mutants has shown that nucleus-wide partitioning of both phosphorylated HIM-3 as well as phosphorylated SYP-1 correlates with the number of crossover intermediates. In addition, irrespective of whether partitioning occurs, loss of phosphorylated SYP-1 from chromosomes without COSA-1 precedes loss of unphosphorylated SYP-1, suggesting that phosphorylated SYP-1 has higher affinity toward chromosomes with crossover intermediates. Two possibilities can be considered for the lack of partitioning in nuclei with low numbers of COSA-1 foci: either crossover intermediates generate a signal that must achieve a certain threshold to enable partitioning, or a factor that is normally distributed between 6 chromosomes "spills over" and can no longer enable partitioning when the number of crossover intermediates is low. While we expect further genetic studies to illuminate this question, at this moment we can say that the condition of having a sub-threshold number of crossover designations affects both chromosome-level (SC dissociation from non-crossover chromosomes) and nucleus-level (SC partitioning to short arms) phenomena. Previous studies have proposed that the *C. elegans* SC goes through two distinct states to mature: an early dynamic state prior to CO designation, in which SC central proteins rapidly bind to and dissociate from the SC, and a late, stabilized state of the SC induced by CO-intermediates, in which central proteins are more stably bound to the SC [46,47]. Asymmetric enrichment of phosphorylated SYP-1, HIM-3, and other proteins is likely to be entwined with this stabilized state of the SC, and yet the molecular mechanisms underlying both partitioning and stabilization remain unknown.

We found that HIM-3 phosphorylation is dependent on synapsis, raising the possibility of HIM-3 being phosphorylated by PLK-2, whose localization to meiotic chromosomes depends

on SYP-1 phosphorylation. However, we still observe phosphorylation of HIM-3 in *plk-1 (RNAi)*; *plk-2* mutants, and Ser277 and Ser282 are poor matches for a Polo kinase consensus motif. Further investigation is required to identify kinases phosphorylating HIM-3 at these serines as well as at Tyr279. The synapsis-dependent phosphorylation of HIM-3 we observe is the opposite of what has been shown for mouse HORMAD1 phosphorylation at its closure motif, which is detected only on unsynapsed chromosomes [33], suggesting the possibility of a different mode of axis phosphoregulation. Although phosphorylation-mediated regulation of interactions between chromosome axis components may be conserved at the closure motif among HORMA domain proteins, its use is likely to have diverged in order to adapt to different requirements during meiosis between *C. elegans* with holocentric chromosomes and organisms with monocentric chromosomes. Further investigation is needed to determine how post-translational modification of the chromosome axis and SC central elements contribute to their partitioning, function, and disassembly.

## Materials and methods

List of strains, antibodies, oligos, kits and softwares used in this paper is provided in S2 Table. All the numerical values for the graphs in the Figs listed below are provided in S3 Table: Figs 1C, 5C and 6C, S4, S8, S11, and S14 Figs.

### Strains

*C. elegans s*trains were grown with standard procedures [73] at 20˚C. Wild-type worms were from the N2 Bristol strain. Mutations, transgenes and balancers used in this study are listed in S2 Table. For all mutant analyses, we used homozygous mutant progeny of heterozygous parents.

The following point mutants were generated by CRISPR-Cas9 gene editing to create point mutations at the endogenous *him-3* locus: *him-3(S277A), (S282A), (AA*: *S277A S282A) (FA*: *Y279F S282A), (AFA*: *S277A, Y279F, S282A), (S282D)*. In addition, the following mutants were generated in the *htp-1*::*flag* background to enhance HTP-1 visualization and quantification: *(S282A), (AFA*: *S277A, Y279F, S282A)* and *(S282E)*. For CRISPR-Cas9 gene editing of *him-3* and *syp-1*, gRNAs and synthetic oligonucleotides used as ssDNA homology templates are listed in S2 Table. CRISPR-Cas9 gene editing was done essentially as described in [15] using *dpy-10* as a co-conversion marker [74] with the following modifications: we used crRNA/tracrRNA duplex at 17.5 µM, Cas9 protein at 17.5µM, ssDNA homology template at 6 µM and *dpy-10* ssDNA homology template at 0.5 µM in the injection mix. All ssDNA homology templates, crRNA and trcrRNA were purchased from Integrated DNA technology (IDT) (S2 Table). The *him-3* and *syp-1(T452A)* homology templates include silent mutations (indicated in lower case in S2 Table) creating XbaI or XspI/MaeI/BfaI cut site, respectively to select for homologous recombination products. Genotyping of *syp-1(T452A)* or *him-3* phospho-mutants was done using the primers listed in S2 Table.

To generate the HTP-1::FLAG fusion, C-terminal tagging of the endogenous *htp-1* locus was performed by injecting the germline with a mix of three plasmids according to [75]. The mix contained: 225 ng/ul of a vector carrying sgRNA htp-1Cterminus2 (TATTTACAGGAAC-GAGAATA) driven by the U6 promoter, 75 ng/ul of a vector expressing Cas-9 under the *eft-3* promoter, and 150 ng/ul of vector pCF104 carrying an mCherry reporter and a FLAG tag flanked on both sides by 0.8 kb of sequence homologous to regions upstream and downstream of the Cas9-induced cleavage site at the *htp-1* locus. Selection for homozygous insertion resulted in strain ATG285 *htp-1(fq24[htp-1::FLAG])* IV.

To carry out *plk-1* RNAi in *plk-2(ok1936)* mutants, the *plk-1* gene was amplified from N2 genomic DNA using the following primers: plk-1_RNAi_fw primer (cgtggctagcTCAACAA-CAAGCTGCAGAGG) and plk-1_RNAi_rev primer (catggctagctgggactaaaagggtcgatg). The amplicon was cloned into vector L4440 using NheI restriction enzyme digestion, and this plasmid was used to transform HT115 bacteria. For RNAi, first the *plk-1* RNAi plasmid or L4440 (empty vector) were inoculated into LB+100μg/ml carbenicillin overnight at 37˚C, concentrated 20 times by centrifugation and resuspension, and spread onto NGM plates containing 100 μg/ml carbenicillin and 1 mM IPTG. Next, *plk-2(ok1936)/hT2[bli-4(e937) let-?(q782) qIs48]* heterozygous worms at the larval L4 stage were placed on either *plk-1* RNAi or control plates for three days. Their *plk-2(ok1936)* homozygous F1 progeny were transferred to fresh RNAi plates at both early larval stage as well as L4 stage, and dissected one day post L4 for immunofluorescence.

## Phosphoproteomics

Mass spectrometry of phosphoproteins was as described in [20]. Wild-type N2 and *pph-4.1 (tm1598)/hT2[bli-4(e937) let-?(q782) qIs48]* worms were grown on NGM plates containing 25 μg/ml carbenicillin and 1 mM IPTG spread with HT115 bacteria either carrying an empty RNAi vector (L4440; http://www.addgene.org/1654) or a *pph-4.1*-RNAi plasmid. To generate the *pph-4.1* RNAi plasmid, the 562-bp region spanning the second, third, and fourth exons of pph-4.1 was amplified from *C. elegans* N2 cDNA library using primers 5′-GCT CGT GAA ATC CTA GC-3′ (forward) and 5′-CGA ATA GAT AAC CGG CTC-3′ (reverse) flanked by Not1 and Nco1 sites and cloned into L4440. First, N2 and *pph-4.1(tm1598)/hT2[bli-4(e937) let-?(q782) qIs48]* worms (P0) synchronized by starvation were transferred to new plates with food, and worms at the L4 larval stage were harvested in M9 + 0.01% Tween buffer, washed three times with M9 + 0.01%Tween buffer, and distributed to either control or *pph-4.1* RNAi plates. Approximately 30 h later, these worms on RNAi plates were harvested in M9 + 0.01% Tween buffer and bleached to collect embryos. Collected F1 embryos were distributed to fresh RNAi plates. At time points when these F1 worms were either 1 or 3 d after L4 stage, half of the F1 plates were exposed to 10 Gy γ-rays to induce DNA damage. 4 h after irradiation, worms were harvested in M9 buffer, washed three times with M9 buffer, and frozen at −80˚C. 2 ml pelleted, frozen worms prepared in this manner was thawed and dissolved in 5 ml urea lysis buffer (20 mM Hepes, pH 8.0, 9 M urea, 1 mM sodium orthovanadate, 2.5 mM sodium pyrophosphate, and 1 mM β-glycerol-phosphate), sonicated for 1 min at 30-s intervals 10 times until worm bodies were broken up. The worm lysates were spun down at 20,000 g for 15 min, and supernatants were subjected to PTMScan analysis (Cell Signaling Technology); phosphorylated peptides were enriched by phospho-(Ser/Thr) kinase substrate antibody-immobilized protein A beads and analyzed by liquid chromatography–tandem mass spectrometry using an LTQ-Orbitrap-Elite ESI-CID (Thermo Fisher). Phosphoenrichment antibodies were obtained from CST (catalog numbers 9607, 6966, 8139, 8738, 9624, 6967, 5759, 9942, 10001, 9614, 9477, 8134, 2325, 5243, and 3004). Protein assignments were made using Sorcerer. Peptide counts indicated in **S1 Table** show pooled counts from all conditions of worms (+-RNAi, irradiation, or age) used in this assay.

## Microscopy, cytology, antibodies

For all cytological preparations, we essentially followed protocols described in [76] with the following modifications described below. First, age-matched hermaphrodites were dissected on a coverslip in a 15μl drop of EBT buffer (HEPES pH7.4 27.5mM, NaCl 129.8mM, KCl 52.8mM, EDTA 2.2mM, EGTA 0.55mM, Tween 1%, Tricaine 0.15% weight per volume,

Levamisole 0.015% weight per volume) and fixed by adding 15μl of Fix solution (HEPES pH7.4 25mM, NaCl 0.118M, KCl 48mM, EDTA 2mM, EGTA 0.5mM, Formaldehyde 0.2%) into the drop. Worm fixation was limited to 2 minutes to improve the SC staining. Dissected worms were sandwiched between the coverslip and slide glass (Matsunami Inc. cat # S9901), and were frozen at -80 degree freezer for at least 10 minutes. The coverslip was flicked off using a razor blade, and worms on the slide glass were fixed in -20°C methanol for 1 min and then washed in PBST (PBS + 0.5% Tween) for 10 minutes three times. Slides were blocked in the blocking buffer (PBST + 0.5% bovine serum albumin (BSA)) for 30 minutes and incubated with primary antibodies overnight at 4°C or at least for two hours at room temperature. Then slides were washed in PBST for 10 minutes three times, and incubated with secondary antibodies for two hours at room temperature. Slides were washed once in PBST for 10 minutes and stained with 0.5μg/ml of 4′,6-diamidino-2-phenylindole (DAPI) in PBST for 10 minutes and washed once in PBST for 10 minutes. Finally slides were mounted with 15μl of mounting medium. Mounting medium was made by combining 50μl of 5M Tris (not pH-adjusted) and 450μl of N-propyl gallate (NPG)-glycerol stock (dissolve 2g of NPG in 50ml of glycerol to make NPG-glycerol stock). Worms on the slides were covered by micro cover glass (Matsunami Inc. Cat# No.1-S) and sealed with nail polish. All immunofluorescence was performed on adult worms at 1 day post -L4 unless otherwise mentioned. Depletion of GSP-1 and GSP-2 via AID-mediated degradation was conducted as in [23]. All the primary antibodies and dilutions used in this study are listed in S2 Table. Secondary antibodies (Alexa-488, Dylight-594, Dylight-649) were purchased from Jackson ImmunoResearch Inc. and used at 1:500 dilution for all the immunostaining.

For HIM-3phos antibody production, synthetic HIM-3 peptides (NH2-C+RISRD(pS) PYGL(pS)QGITK-COOH) with phosphorylation at Serine 277 and Serine 282 were synthesized and used to immunize a guinea pig by Eurofins Japan Inc.. This dual phosphopeptide was used as the antigen since it was most abundantly found in our mass spectrometry data (see S1 Table). The antiserum was purified according to manufacturer's instructions using both unphosphorylated (NH2-C+RISRDSPYGLSQGITK-COOH, synthesized by Eurofins Japan Inc.) and phosphorylated HIM-3 peptides (NH2-C+RISRD(pS)PYGL(pS)QGITK-COOH, synthesized by Scrum Inc.) with Sulfolink Immobilization kit for peptides (Thermo Scientific Inc.: cat#44999). Briefly, the antibodies bound to the purification column with phosphorylated peptides were collected, and they were further run through the purification column with unphosphorylated peptides, and the flowthrough fraction, which did not bind to unphosphorylated peptides, was collected as HIM-3phos specific antibodies. Antibody specificity was validated by staining *him-3(S277A S282A)* double mutants (S1 Fig). This polyclonal HIM-3phos antibody is a mixture of antibodies recognizing S277 phosphorylation as well as antibodies recognizing S282 phosphorylation since we found immunofluorescence staining both in *him-3(S277A)* single and *him-3(S282A)* single mutants using this antibody (S2 Fig and S3 Figs).

For SYP-2 antibody production, SYP-2 antigen peptides (NH2-C +AHYDKLLDLVETLEPWADKL-COOH) were synthesized and used to immunize rats by Eurofins Japan Inc.. The antiserum was purified according to manufacturer's instructions with the antigen peptides using Sulfolink Immobilization kit for peptides (Thermo Scientific Inc.: cat#44999).

Imaging was performed on slides prepared for immunofluorescence on a Deltavision microscope system, using a 100x, 1.4NA PlanSApo objective (Olympus Inc.). Images were taken on a Photometrics CCD camera (1024x1024 pixels) at an effective pixel size in the image plane of 64 nanometers. After collecting images, raw 3D data was corrected for lamp flicker and Z-dependent bleaching, then deconvolved using a measured point spread function using

the softWoRx suite (GE Healthcare Inc.). After deconvolution, wavelengths were offset in the Z direction to correct for chromatic aberration, using multicolor beads as a calibration standard. Quantification of apoptotic nuclei was carried out as described in [63]. For quantification of SYP-1 and HTP-1 localization patterns, oocyte precursor cells from the -4 through -7 diakinesis stage were scored. For quantification of ICP-1 localization pattern, nuclei from -3 through -1 diakinesis stage were scored. For visualizing GFP::COSA-1, immunostaining against GFP was used.

For quantitative analysis of HIM-3 and HIM-3phos intensities (Fig 3), late pachytene nuclei in raw (non-deconvolved) 3D images were classified into "pre" or "post" based on partitioning of HIM-3phos or SYP-2 protein. Ten nuclei of each class were selected per gonad. Within each nucleus, 50 points along the chromosome axis were picked semi-manually in the HIM-3phos channel using the PickPoints feature of the Priism software suite [77]. Points were automatically selected as the maximum intensity pixel within a 5x5x3-pixel box in X, Y, and Z centered on the clicked point, and constrained to ensure points were not duplicated. To compare intensities of pan-HIM-3 on short and long arms, the same nuclei from the "post partitioning" zone were used and an additional 50 points were picked in each nucleus in the pan-HIM-3 channel on chromosome stretches where HIM-3-phos points had not been picked previously. To compare intensities between different datasets, points were normalized by dividing all intensity values by the mean intensity of the "pre partitioning" zone points. Significance testing was performed in R 3.6.1 (function t.test).

For HTP-1::FLAG intensity measurements, Fiji [78] was used to calculate the intensity of FLAG antibody staining in control *(htp-1::flag)* and mutant animals (*him-3(S282A or S282E) htp-1::flag*), dissected and stained on the same slides. First, raw (non-deconvolved) 3D images were sum-projected, and regions of interest were drawn based on HTP-1::FLAG positive pixels in −1 through -3 oocyte nuclei, and the total FLAG pixel intensity within regions of interest was measured. To subtract background intensity, two independent background regions were drawn in the same nucleus outside of the chromosomes, whose mean pixel intensity was taken as the background intensity. Mean background intensity value was multiplied by the region of interest's area to calculate the total background intensity, which was subtracted from the total FLAG intensity value. Intensity data were statistically analyzed by the Mann–Whitney U-test.

## Embryonic viability quantification

To score embryonic viability of self-fertilized worms, P0 animals at the L4 stage were transferred to new plates every 24 hours. Eggs remaining on the plates 20–24 hours after the transfer were scored as dead eggs. The number of hermaphrodite or male progeny reaching L4/adult stages were scored three days after the transfer. For the γ-irradiation experiment, P0 worms at the L4 stage were exposed for 87 minutes at 0.855 Gy/min (total exposure 75Gy) in a $^{137}$Cs Gammacell 40 Exactor (MDS Nordion) and scored for their embryonic viability as above.

## Fluorescence Polarization (FP) anisotropy peptide assay

Fluorescein (FITC) conjugated peptides for the FP assay were synthesized by either Scrum Inc. or Eurofins Inc. as shown in **S2 Table**. HTP-2 proteins were purified and FP assays were conducted as described in [27]. For the fluorescence assay, synthesized peptides were first resuspended in DMSO, then diluted into binding buffer (20 mM Tris pH 7.5, 300 mM NaCl, 10% glycerol, 1 mM DTT, 0.1% NP-40). Reactions of 50 μl total volume containing 50 nM peptide and various concentrations of HTP-2 proteins were incubated for 1 hour at room temperature, then fluorescence polarization was measured in 384-well plates using a TECAN Infinite M1000 PRO fluorescence plate reader. Binding data were analyzed with Graphpad Prism

version 6.0h for Macintosh (GraphPad Software, La Jolla California USA) using a single-site binding model.

## Western blot

Two hundred adult hermaphrodites were picked into M9 buffer (41 mM Na2HPO4, 15 mM KH2PO4, 8.6 mM NaCl, 19 mM NH4Cl), washed with M9 buffer for three times, and boiled for 10 minutes in SDS-PAGE sample buffer. SDS-PAGE was carried out using precast 10% TGX gels (Biorad Inc.), and proteins were transferred to a PVDF membrane. The membrane was blocked with TBST buffer (TBS and 0.1% Tween) containing 1% BSA at 4˚C overnight and probed with panHIM-3 antibodies (Novus Inc., Cat #53470002, 1/3000 dilution in TBST) or anti-tubulin antibodies (Santa Cruz Inc. Cat# SC32293, 1/4000 dilution) for 2 hours at room temperature, washed with TBST for three times, probed with either anti-mouse-HRP (Amersham Inc. Cat#NIF825, 1/20000 dilution in TBST) or anti-rabbit-HRP antibodies (Abcam Inc., Cat # ab6721, 1/20000 dilution in TBST) for two hours, washed with TBST for three times. Chemiluminol assay kit, Chemilumi-one super (Nacalai Inc.), was used to visualize protein bands. Band quantitation was performed with Fiji [78].

## Supporting information

**S1 Table. Counts of phosphopeptides obtained from mass spectroscopy.** Details of phosphopeptides detected are shown. For all phosphopeptides detected only once, mass spectrometry spectra were manually examined, and peptides with high confidence assignments are indicated. The MS/MS spectra showing m/z and intensity values for these manually reviewed peptides are shown with accompanying spectrum images in sheets 2 through 4 in the same Excel file.
(XLSX)

**S2 Table. List of antibodies, worm strains, oligos, kits and softwares used in this paper.**
(XLSX)

**S3 Table. Numerical values for the graphs presented in this paper.**
(XLSX)

**S1 Fig. A**, Diagrams of *C. elegans* HIM-3 (top) and mouse HORMAD1 (bottom), showing the positions of the closure motif and phosphoserine at the C-terminus. **B**, Antibodies specific to the phosphorylated closure motif stain the meiotic chromosome axis in wild-type (*top*, green in merged image) but not in S277A,S282A mutants (*bottom*), whereas pan-HIM-3 antibodies stain both (magenta in merged images), verifying the antibody's specificity to the phosphoepitope. Scale bars, 5μm. **C**, Immunostaining of panHIM-3 and HIM-3phos in the N2 wild type gonad to show the progression of HIM-3phos enrichment to confinement. While HIM-3phos signals are mostly enriched on short arms from late pachytene, faint HIM-3phos staining, indicated by an arrow, is still detectable on long arms in early diakinesis and then disappears from long arms by -1 diakinesis. Scale bars, 5μm in the left panels and 1μm in the right panels. **D**. Immunostaining of HIM-3phos, panHIM-3 and panSYP-1 from late pachytene through diplotene to show temporal progression of HIM-3phos and panSYP-1 enrichment on short arms. Scale bars, 5μm.
(TIF)

**S2 Fig. A,** Immunostaining of panHIM-3 and HIM-3phos (detecting specifically S282 phosphorylation) antibodies in the *him-3(S277A)* gonad. Scale bars, 5μm. **B**. Immunostaining of HIM-3phos (green in merged image), SYP-1, and pan-HIM-3 (magenta in merged image)

antibodies in *him-3(S277A)* mutants. HIM-3phos (Ser282phos) staining is confined to short arms from late pachytene onward. Scale bars, 5μm, and 1μm in the magnified bivalent image. (TIF)

**S3 Fig. A,** Immunostaining of panHIM-3 and HIM-3phos (detecting specifically S277 phosphorylation) antibodies in the *him-3(S282A) htp-1::flag* gonad. Scale bars, 5μm. **B**. Immunostaining of HIM-3phos (green in merged image), SYP-2, and pan-HIM-3 (magenta in merged image) antibodies in *him-3(S282A)* mutants. HIM-3phos (S277phos) staining remains on both short and long arms until -1 diakinesis and does not become enriched on short arms in *him-3 (S282A) htp-1::flag* mutants. Scale bars, 5μm, and 1μm in the magnified bivalent image. (TIF)

**S4 Fig. A**, **B, C**: Individual points plotted for each nucleus within each of the 6 gonads scored for HIM-3 intensity quantitation in **Fig 3**. Each point represents the mean normalized intensity value for 50 pixels picked in one nucleus for the given class. **A,B**: Comparison of the levels of phosphorylated HIM-3 (**A**) or panHIM-3 (**B**) on short arms after partitioning with that of the entire chromosome axis before partitioning within the same gonad. **C**, Comparison of the levels of panHIM-3 on short arms versus long arms in post-partitioned nuclei. Error bars within each group indicate 95% confidence intervals. "Pre" means pre-partitioned nuclei, "post" means post-partitioned nuclei. Statistical significance was tested by two-tailed *t*-tests between both conditions in each gonad; p value upper bounds are indicated for each comparison; no value indicates p > 0.05. **D**, GSP-1/2 are not essential for asymmetric distribution of phosphorylated HIM-3. Immunostaining against phospho-HIM-3 (green in merged image) and pan-HIM-3 (magenta in merged image) in the control (N2 + auxin) and GSP-1/2-depleted gonad *(gsp-1(fq51 [gsp-1::degron]); gsp-2 (fq49 [gsp-2::degron]); ieSi38 [Psun-1:: TIR-1::mRuby:: sun-1 3'UTR + Cbr-unc-119(+)] + Auxin)* as well as in a *gsp-2(tm301)* null mutant. HIM-3phos staining still disappears from long arms and becomes enriched on short arms in the absence of GSP-1/2. **E**, Immunostaining of HIM-3phos (green in the merged image), SYP-1 and pan-HIM-3(magenta in the merged image) in *him-3(S277A)* (top) and *him-3(S277A); atm-1 (gk186); atl-1(tm853)* (bottom) oocyte precursor cells at late pachytene. Although many polyploid nuclei were seen in the *atm-1(gk186); atl-1(tm853)* background, SC structures were partially formed, and HIM-3 within these structures was still phosphorylated. Scale bars, 5μm. (TIF)

**S5 Fig. HIM-3 phosphorylation depends on synapsis, but not Polo-like kinases or CHK-2 kinase. A**, Immunostaining of phospho-HIM-3 (green in merged image), pan-HIM-3 (magenta in merged image) and SYP-2 in the indicated genotypes. In the *plk-2(ok1936)* mutant background (bottom row), arrowheads point to chromosomes that lack synapsis (by α-SYP-2 immunostaining); these same chromosomes also lack HIM-3phos but not pan-HIM-3. **B,** Immunostaining of phospho-HIM-3 (green in merged image), pan-HIM-3 and SYP-2 (magenta in merged image) in *control RNAi (L4440); plk-2 (ok1936)* or *plk-1 (RNAi); plk-2 (ok1936)* mutants. HIM-3phos staining is detected on synapsed chromosomes in *plk-1 (RNAi); plk-2(ok1936)* mutants. **C**. Immunostaining of HIM-3phos (green in merged image), SYP-1phos (magenta in merged image) and HTP-3 in the *chk-2(me64) rol-9(sc148)* mutant and N2 wild type. To identify *chk-2(me64) rol-9(sc148)* homozygous animals, 72 hours post L4 stage adult worms were dissected after verifying the presence of dead eggs due to *chk-2 (me64)* homozygosity. The *rol-9* (sc148) is used to mark *chk-2(me64) in cis*, and it not expected to cause any meiotic phenotype. HIM-3phos staining is detected on synapsed chromosomes in *chk-2(me64) rol-9(sc148)* mutants. Scale bars, 5μm. (TIF)

**S6 Fig.** **A**, Immunostaining of GFP (green in merged image), HIM-3phos and panHIM-3 in late pachytene in *spo-11(me44); gfp*::*cosa-1* mutants. Meiocytes with GFP::COSA-1 are circled with dotted lines and show enrichment of HIM-3phos signals on the chromosomes with GFP::COSA-1. **B,** Embryonic viability, male progeny production indicating the rate of *X* chromosome nondisjunction, and total number of scored embryos is shown for the indicated genotypes. p values indicated are from chi-squared test for independence of the embryonic inviability or male counts compared to N2, with Bonferroni correction applied. **C,** Western blot of control (anti-tubulin) and pan-HIM-3 for adult worms with indicated genotype. Similar levels of HIM-3 proteins are detected in *him-3* phospho-mutants compared to control animals. Scale bars, 5μm.
(TIF)

**S7 Fig. A,** Immunostaining of panHIM-3 in pachytene nuclei in the genotypes indicated. The levels and localization of pan-HIM-3 staining in *him-3* phospho-mutants appears comparable to that of control gonads. **B**, Immunostaining with anti-FLAG epitope or HTP-1 (magenta in the merged image) and SYP-1(green in the merged image) in the leptotene/zygotene transition zone in the gonads with indicated genotype. Normal loading of HTP-1 and SYP-1 is observed in *him-3* phospho-mutants. Arrows show the direction of meiotic progression in each gonad. Scale bars, 5μm.
(TIF)

**S8 Fig. A and B,** Immunostaining with anti-FLAG epitope on germlines expressing a transgenic HTP-1::FLAG fusion protein in an otherwise wild-type background or a *him-3(S282E)* mutant background at diakinesis (A), or in an wild-type background or a *him-3(S282A)* mutant background (B) stained on the same slide, acquired under the same imaging conditions, and displayed with the same scaling. Images shown are summed projections of raw (not deconvolved) image data. Scale bars, 5μm. Quantitation of the intensity of FLAG::HTP-1 immunostaining in -3 through -1 diakinesis nuclei between the two genotypes in A or B are shown in the right panel. (Significance tested with unpaired two-tailed t-test after passing Shapiro-Wilk normality test, p<0.0001 for A and p<0.001 for B) Scale bars, 5μm.
(TIF)

**S9 Fig. Persisting SYP-1 eventually dissociates from both arms at the normal timing (-3/-2 diakinesis) in him-3 phospho-mutants, and persisting HTP-1 eventually dissociates from short arms in him-3(S282E) mutants.** Immunostaining of panHIM-3 and either SYP-1 or FLAG in control *(htp-1::flag)*, *him-3(S282A) htp-1::flag* and *him-3(S282E) htp-1::flag* mutant gonads. The same SYP-1 panels of *him-3(S282A) htp-1::flag* and *him-3(S282E) htp-1::flag* mutants as well as the HTP-1 panel of *him-3(S282E) htp-1::flag* mutant are presented in the main **Fig 6**. Scale bars, 5μm.
(TIF)

**S10 Fig. The timing of OMA-2 expression, a marker for maturing oocytes, is normal in him-3 phospho-mutants.** Immunostaining of OMA-2 in *htp-1*::*flag control*, *him-3(S282E) htp-1*::*flag* and *him-3(S282A) htp-1*::*flag* mutant gonads. DAPI is shown in green in merged images. Scale bars, 5μm.
(TIF)

**S11 Fig. Immunostaining of CPC component INCENPICP-1 (green in merged images) and HTP-3 (magenta in merged images) on individual diakinesis chromosomes in *N2* (wild type), *him-3(S282E) htp-1::flag, him-3(S282D), him-3(AFA) htp-1::flag* and *him-3 (S282A) htp-1::flag*.** Shown are partial maximum-intensity projections of single chromosomes

oriented so the long arms are vertical. Scale bars, 1μm. Quantification of ICP-1 localization classes (confined or unconfined) is shown at right, along with numbers of nuclei scored. Data shown are from at least 2 independent biological replicates. p-value of Fisher's exact test for independence after correction for multiple comparisons is shown.
(TIF)

**S12 Fig. A**, Asymmetric enrichment of phosphorylated HIM-3 is abrogated in *syp-1(T452A)* or *plk-2(ok1936)* mutants. Immunostaining of phospho-HIM-3 (green in merged images) and pan-HIM-3 (magenta in merged images) in the indicated genotypes showing phospho-HIM-3 failing to partition the short arm. Partial Z projections are shown for all images. **B,** Phosphorylated HIM-3 colocalizes with SYP-2 along the entire SC in late pachytene, and to presumptive CO designation sites in diplotene in *plk-2 (ok1936)* mutant gonads. Scale bars, 5μm, and 1μm for the magnified bivalent images.
(TIF)

**S13 Fig. A**, Immunostaining of phospho-SYP-1 (green in merged images) and HTP-3 (magenta in merged images) in the indicated genotypes. For both A and B, individual diakinesis chromosomes displayed are circled in the overview images. In *him-3(S282E)* or *(AFA)* mutants, although SYP-1phos staining is confined normally to short arms from late pachytene up to -5 diakinesis, SYP-1phos signals reappearing on long arms are occasionally found specifically in -4 and -3 diakinesis nuclei, presumably due to *de novo* phosphorylation activity by yet-unidentified kinase in mature oocytes. Scale bars: individual diakinesis chromosomes, 1μm; all others 5μm. **B and C**, Percentage of embryonic viability among the self-progeny of worms with the indicated genotypes. Total number of eggs scored are: *him-3(S282A)*: 1583, *him-3(Y279F S282A)*: 1613, *syp-1(T452A)*1129, *syp-1(T452A); him-3(S282A)*: 1449, *syp-1 (T452A); him-3(Y279F S282A)*: 1336 *N2*: 1494 in graph B, and *syp-1(T452A)*: *1413, syp-1 (T452A); him-3(S282E) htp-1::flag*: 1371, *him-3(S282E) htp-1::flag*:1218 in graph C. Mean with 95% confidence intervals is indicated in the graph. No statistical significance was detected between the *syp-1(T452A)* single mutant with *syp-1(T452A); him-3* (S282A or Y279F S282A or S282E) double mutants using Kruskal-Wallis analysis with Dunn's multiple comparison test.
(TIF)

**S14 Fig. A**, Percentage of embryonic viability among the self-progeny of worms with the indicated genotypes. For each column, n indicates the number of maternal worms. Total number of eggs scored are: *him-3(S282A)*:1185, *him-3(AFA)*: 1370, *brc-1(tm1145) brd-1(dw1)*: 3013, *him-3(S282A); brc-1(tm1145) brd-1(dw1)*: 2178, *him-3(AFA); brc-1 brd-1*: 2443 eggs. Mean with 95% confidence intervals is indicated in the graph. The *him-3* non-phosphorylatable mutations did not lower embryonic viability when sister-chromatid-mediated homologous recombination is prevented in the *brc-1(tm1145) brd-1(dw1)* background. **B**, Percentage of embryonic viability among the self-progeny of P0 worms after 75 gray γ-irradiation at L4 stage. The *brc-1(tm1145) brd-1(dw1)* mutant was used as a positive control, which shows reduced embryonic viability upon gamma irradiation. Total number of eggs scored are: *brc-1 (tm1145) brd-1(dw1)*: 646, *htp-1::flag*: 1178, *him-3(AFA) htp-1::flag*: 1896, *him-3(S282E) htp-1:: flag*: 1233 eggs. Mean with 95% confidence intervals is indicated in the graph. The *him-3* phospho-mutations did not lower embryonic viability upon exogenous DNA damage using Kruskal-Wallis test with Dunn's multiple comparison. **C**. Scoring of apoptotic nuclei in *him-3* phospho-mutant backgrounds. The number of CED-1:GFP-positive nuclei for each condition is plotted with mean and 95% confidence intervals. The *him-3* non-phosphorylatable mutations did not increase the number of apoptotic nuclei.
(TIF)

**S15 Fig. Schematic diagrams of observed phenotypes in wild type and *him-3* phospho-mutant animals.**
(TIF)

## Acknowledgments

We thank A.F. Dernburg, A. Villeneuve, H. Kawahara and K. Oegema for antibodies and the past and current members of the Carlton lab for technical assistance. Many nematode strains were provided by the Caenorhabditis Genetics Center, which is funded by the National Institutes of Health National Center for Research Resources.

## Author Contributions

**Conceptualization:** Aya Sato-Carlton, Kevin D. Corbett, Peter Mark Carlton.

**Data curation:** Peter Mark Carlton.

**Funding acquisition:** Aya Sato-Carlton, Enrique Martinez-Perez, Kevin D. Corbett, Peter Mark Carlton.

**Investigation:** Aya Sato-Carlton, Chihiro Nakamura-Tabuchi, Xuan Li, Hendrik Boog, Madison K. Lehmer, Scott C. Rosenberg, Consuelo Barroso, Peter Mark Carlton.

**Methodology:** Aya Sato-Carlton, Madison K. Lehmer, Scott C. Rosenberg, Kevin D. Corbett.

**Project administration:** Aya Sato-Carlton, Peter Mark Carlton.

**Resources:** Consuelo Barroso, Enrique Martinez-Perez.

**Supervision:** Aya Sato-Carlton, Peter Mark Carlton.

**Visualization:** Aya Sato-Carlton, Madison K. Lehmer, Kevin D. Corbett, Peter Mark Carlton.

**Writing – original draft:** Aya Sato-Carlton, Peter Mark Carlton.

**Writing – review & editing:** Aya Sato-Carlton, Enrique Martinez-Perez, Kevin D. Corbett, Peter Mark Carlton.

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
