## [Decision Letter · Decision Letter 0]

22 Jul 2020

Dear Dr Carlton,

Thank you very much for submitting your Research Article entitled 'Phosphoregulation of HORMA domain protein HIM-3 promotes asymmetric synaptonemal complex disassembly in meiotic prophase in C. elegans' to PLOS Genetics. Your manuscript was fully evaluated at the editorial level and by three independent peer reviewers. The reviewers appreciated the attention to an important problem, but raised some substantial concerns about the current manuscript. Based on the reviews, we will not be able to accept this version of the manuscript, but we would be willing to review a much-revised version. We cannot, of course, promise publication at that time.

Should you decide to revise the manuscript this should include, but not be restricted to, the following: 1) completing the analysis of known C. elegans meiotic kinases- such as by testing HIM-3 phosphorylation in plk-2; plk-1 double mutants, 2) testing the model of redundancy of establishment of short and long arms by the analysis of the appropriate double mutant, 3) The reviewers also point to some cases in which data is not shown but described, validation of reagents that is missing, and insufficient experimental detail description- please provide all the data stated but not shown in the supporting material and the missing experimental details and reagent validation in the methods section, 4) all data quantified should have the appropriate statistics clearly stated: significant p values indicated on figures, and a section in materials and methods that will describe the statistical analysis (e.g., which normality test was used before t-test was performed?) for all quantified data set, including stacked bar graph data and data in tables.

In addition I have the following critiques as editor that I would like you to respond to:

• The image emerging from the analysis of the HIM-3 mutants is complex. Adding a Sup figure explaining the author’s model and the mutants’ phenotypes will be very helpful for the reader.

• S4A- Why does the htp-1 flag tag in him-3(S282E) background show emb, but not on its own? Also, in S7b the flag tag shows emb after IR (assume more the WT, but there is no control). These 2 observations imply that the tag is not fully functional.

• S7A and B- Wild type control is missing.

• S7A -Why does the htp-1::flag (on its own) show emb after IR and why is it rescued by the mutants?

• S4- Shows just S282E, there is no data showing the effect on HTP localizations level in other mutants.

• Images of FAF mutants 6B shows axis/bivalents partially detaching- is this a common phenotype?

• Based on the text of figure 6, the misloclization of HTP-1 and SYP-1 in the HIM-3 mutants is eventually resolved. The authors need to present the data supporting this statement in figure 6 (not sup).

• Typos:

o 1um labeling in 6B is misplaced.

o S7b is flag::htp-1 supposed to be htp-1::flag?

o Fig 4B, 5B scale bar moved up in the bottom panels.

o S3- italics where it should not be needed on labels.

If you decide to revise the manuscript for further consideration at PLOS Genetics, please aim to resubmit within the next 60 days, unless it will take extra time to address the concerns of the reviewers, in which case we would appreciate an expected resubmission date by email to plosgenetics@plos.org.

[LINK]

We are sorry that we cannot be more positive about your manuscript at this stage. Please do not hesitate to contact us if you have any concerns or questions.

Yours sincerely,

Sarit Smolikove

Guest Editor

PLOS Genetics

Gregory P. Copenhaver

Editor-in-Chief

PLOS Genetics

Reviewer's Responses to Questions

**Comments to the Authors:**

Reviewer #1: This manuscript “Phosphoregulation of HORMA domain protein HIM-3 promotes asymmetric synaptonemal complex disassembly in meiotic prophase in C. elegans” presents insights into the role of phosphorylation in regulating meiotic chromosome remodeling and SC disassembly. Meiotic HORMA domain proteins in C. elegans, HIM-3, HTP-1, -2, and -3, form hierarchical assemblies through binding of their HORMA domains to cognate peptide motifs termed “closure motifs” and play essential roles during meiosis. Here, the authors identified that HIM-3 is phosphorylated at its closure motif, thereby disrupting its binding to HTP-1/2 in vitro. Using phospho-specific antibody, the authors showed that HIM-3 phosphorylation occurs along synapsed axes in early meiotic prophase, but becomes enriched on the short arm upon crossover formation. It was further demonstrated that HTP-1 release and SC disassembly are delayed in worms expressing phospho-mimetic mutations of HIM-3, supporting the role of HIM-3 phosphorylation in axis remodeling.

While it is clear that the phosphorylation at S282 dramatically reduces HTP-2 binding in vitro, the results from in vivo analyses using him-3 mutants are somewhat confusing. For example, both him-3(S282E) and him-3(S282D) worms contain phospho-mimetic mutations and display similar defects in SC disassembly (Figure 6C); however, they show distinct phenotypes in ICP-1 distribution (Figure S5). Can the authors explain why this is the case?

Another confusing part (to this reviewer) is in Discussion to explain how the delay in HTP-1 release is observed in him-3 phospho-mimetic mutants, but not in phosphorylation-defective mutants. The authors speculate that, in him-3 phospho-mimetic mutants, HTP-1/2 are likely recruited to the axis through binding HTP-3 motifs, which may result in less efficient dissociation. However, this same scenario will be applied to axis-recruited HTP-1/2 in non-phosphorylatable mutants, where a pool of HTP-1/2 is bound to HTP-3 motifs and another pool is bound by HIM-3. The readers will greatly benefit from some clarification and potentially further experimentation (e.g. what will happen if him-3 phospho-mutations are combined with HTP-3 mutations where HTP-1/2-binding closure motifs are mutated?).

Other major points:

1. Please describe how the HIM-3 phospho-specific antibody was generated and purified and how the phospho-specificity has been validated in Materials and Methods.

2. The phospho-HIM-3 antibody was raised against a phospho-peptide containing phosphates at both S277 and S282. Thus, it is unclear whether the signal reflects the phosphorylation event at both residues or whether it recognizes either pS277 or pS282. This has to be established as an important validation of the antibody specificity. Have the authors tested the phospho-HIM-3 antibody in him-3(S282A) mutants? Typically, a phospho-epitope is very tight around the phosphorylated residue (if purified correctly). It will help interpret the temporal and spatial pattern of HIM-3 phosphorylation at S282, the residue that the authors mostly focused on in this paper.

3. What is the binding affinity of the peptide harboring phospho-mimetic mutations (S282E or S282D) in vitro? It will be important to test whether these mutations truly mimic phosphorylation. It will also help interpret the phenotype seen in worms expressing HIM-3(S282E vs. S282D).

Minor points:

1. The authors showed that HIM-3 phosphorylation depends on the SC formation, but is still present along synapsed axes in plk-2 mutants. How about plk-2; plk-1 double mutants? In C. elegans, PLK-1 is known to substitute for PLK-2 function in the germline.

2. The authors show that partitioning HIM-3 phosphorylation on the short arm depends on SYP-1 T452 phosphorylation. What does the phospho-HIM-3 staining look like after crossover formation in plk-2 mutants? Does it still exhibit short arm enrichment?

3. What would be the significance of HIM-3 phosphorylation along synapsed axes in early meiotic prophase?

Reviewer #2: The manuscript by Sato-Carlton et al., describes the identification and functional characterization of HORMA axis protein, HIM-3 phosphorylation, in C. elegans meiosis. Using a combination of biochemistry, cell biology and genetics, the authors find that 1) HIM-3 is phosphorylated on the conserved C-terminal closure motif, 2) phosphorylation blocks HIM-3 binding to HTP-2 in vitro, leading to a model whereby phosphorylation contributes to removal of HTP-1/2 from the short arm, 3) phosphorylated HIM-3 becomes enriched on the short-arm of the bivalent dependent on the formation and number of crossovers, 4) phosphorylation of HIM-3 plays a role in the timely disassembly of the SC. This is a well-executed study that provides insight into the role and regulation of phosphorylation on SC axis and central region components and illustrates the complex and redundant nature of establishing chromosome domains for accurate chromosome segregation. Please consider the following:

1. The introduction would benefit from some rewriting. Here are a couple of examples/suggestions: page 2: In contrast, organisms with holocentric . . . must define two separate domains . .” (vs. separation) and “At meiosis I, cohesion . . . on long arms persists until degraded . . .(remove is finally). Page 3: The way the ZHP section is written implies that ZHP-1 is a partner with ZHP-3 and ZHP-2 is a partner with ZHP-4; this is incorrect.

2. Figure 2: please indicate in the legend that the cruciform structure observed is dependent on orientation. In 2C, it appears that there is weaker staining of phosphoSYP-1 immediately adjacent to the COSA-1 foci compared to phosphoHIM-3 – is this consistent? Any thoughts?

3. Figure 5: I would like to see a discussion on how the finding that the total number of COs promotes asymmetric distribution of phosphoHIM-3 to the short arm relates to the surveillance pathway that individually monitors crossovers on each chromosome and maintains checkpoint signaling until all chromosomes have a CO as described by Yanowitz and colleagues (Machovina et al., 2016). Did the authors consider treating spo-11 mutants with IR and examining CO numbers and phosphoHIM-3 enrichment to the short arm?

4. Figure 6: Phenotypic consequence of HIM-3 phosphorylation. Does the delay in SYP and HTP-1/2 removal from chromosome domains affect the timing of cohesin removal? Given the subtle phenotypes, analysis of the double syp-1-nonphospho; him-3-nonphospho mutant would be informative. If syp-1-nonphospho mutant is epistatic, it will support the model. Alternatively, if there is any synergism, this will also be a meaningful result. It would also be interesting to combine the him-3 nonphospho mutant with the syp-2 nonphosho mutant as well as combining him-3 nonphospho mutant with HTP-3 mutants that block HTP-1/2 binding. I realize these mutants may not be in hand but what I am getting at is to find a situation where the role of the HIM-3 non-phospho protein is uncovered. This would lend support to the importance of the phosphorylation as well as the redundant nature of defining chromosome domains.

5. Phospho-mimetic mutants: It is surprising that the phospho-mimetic mutants have a more severe phenotype compared to the non-phosphorylated mutants – I would like to see this discussed more and acknowledged that mutation to E or D may not completely mimic the effect of phosphorylation. Are these mutants dominant?

6. Please add p values to Figures where appropriate, e.g., Supplemental Figure 2 and 7B.

Reviewer #3: In this paper Sato-Carlton et al. investigate the post-translational modifications in the meiotic axis HORMA-domain protein HIM-3 and the effects SC restructuring in preparation for the meiotic division. They discovered this modification through Mass Spectrometry and have used a combination of genetics and immunohistochemistry to tease out its role during meiosis. The topic of the study is certainly of interest to the genetics community as post-translational modifications appear to be critical regulators of meiotic processes and our understanding of these is still in its infancy. While the authors present convincing evidence that the phorphorylation of HIM-3 is downstream of SYP-1 and helps in the partitioning of the long and short axes, ultimately, the modification is not essential for the restructuring as these are resolved by the -1 oocytes. Thus, the post-translational modification appears to be adding to the robustness of this process. In this regard, the authors do an excellent job interrogating different models of partitioning and the interplay between the SC proteins and HIM-3 and HTP-1/2. The paper is lacking in two areas: the major phenotypes are in late meiotic events, and while they allude to early defects in meiosis (such as in the syp and plk-2 mutants), the images to evaluate early meiotic events are not provided; and perhaps more importantly, they were unable to identify the relevant kinase—although they only tested a few possible candidates. Thus, while a thorough analysis of new modification, one is left somewhat wanting for more (perhaps an RNAi of known kinases).

While identification of the kinase (or a phosphatase) would tie the paper up in bow, there are a few other issues that need to be clarified prior to publication.

Major Issues:

Figure 1C should have statistical analysis provided. This will show whether the control with Ala is really not significant, as well as to see whether the phosphorylation of S279 is significantly different or not.

One of the problems in analyzing some of the figures is that we see only snapshots of one or two nuclei and not the whole region or the whole meiotic germ line. This is especially true for Figure 4 (spo-11) but also to address their comments that the timing of SC formation is “normal” where the images are not shown to support this statement. These should be provided in the supplemental data.

In Supplemental Figure 1C, HIM-3phospho staining appears more robust in the atm-1;atr-1 double mutants. This is not addressed.

P 13 states “In all of the him-3 mutants, HIM-3 proteins localize to the SC at normal levels by pan-HIM-3 antibody staining” but this is not shown. Also for the next sentences: “In him-3 phosphomimetic

mutants, HTP-1 was detected on the SC from the leptotene/zygotene transition zone,

and SC central elements were polymerized normally.” This is also not shown.

Since the HIM-3phos antibody recognizes the combination of Ser 277 and Ser 282, is it possible that the remaining signal on the long arm represents a partially phosphorylated intermediate (on one of the two sites)?

P 10 and Figure 3A, they say that there is an enrichment on HIM-3 at the “very end of late pachytene, but the images are not those immediately adjacent to diplotene. This should be quantified as the number of rows before the diplotene transition or by the percent of distance along the leptotene- diplotene germ line. This also comes up on p. 11 in discussion of the HIM-3phos staining in spo-11(me44). I think the term “very late pachytene” needs to be defined since clearly there are still adjacent nuclei that are in pachytene and not diplotene.

It would be interesting if the authors say in which stage they observe the enrichment of SYP-1 phosphorylated along the entire length of chromosomes harboring COSA-1 foci in spo-11(me44), because if happens in late stages this conflicts with previous results where a desynapsis is observed in nuclei with subthreshold of CO.

The persistence of HTP-1 on both arms of the bivalent in the him-3 phosphomimetic backgrounds is discussed as a potential “timer” for dissociation. However, it is also possible that there is an overall delay in the events of late pachytene- diakinesis. This could at least be addressed by staining for meiotic maturation markers (e.g. wee-1; oma-1; etc) and making sure they appear with the correct timing.

Minor issues with Figures

Figure 2B does not say that the diakinesis oocyte is the -1 oocyte. Given the discussion of the localization of HIM-3phos in the -1 oocyte, they really should provide images of -1 to -7 oocytes showing the transition.

In the Supplemental Figure 3A, it would be helpful to show again the WT controls.

In the Supplemental Figure 3B where they show that HIM-3phos staining was absent from unsynapsed chromosomes, the nuclei they decided to point out with the arrowheads are not the most obvious (at least not the one on the left). I would suggest selecting others to point to since there are multiple better examples (dead center, e.g.) that would make the point more strongly.

Figure 5C and Figure 6C, Y axes are not labeled

Referencing:

In the first Figure the authors conclude that the examination of the crystal structure led them to predict that phosphorylation at Ser282 would interfere with HTP-1/2 binding. Kim et al., 2014 needs to be referenced here.

The interpretation on p. 12 that recombination intermediates alter the SC in cis was first shown in Machovina et al., 2016 and should be referenced.

References for the function of BRC-1 from Janisw et al 2018 and Li, Saito et al., 2018 should be added on p. 16 when this gene is introduced for its molecular function.

Minor edits:

The paper is extremely well written and a delight to read. The only sentence that was confusing was the top of p. 14 starting: “This implies that although HTP-1 may bind…”. I really have no idea what they are trying to communicate here: are they trying to say that there are 2 pools of HTP-1, one that binds phosphorylated HIM-3 and one that binds unphosphorylated and that these much be mutually exclusive to explain the decrease staining of HTP-1 in the HIM-3D or E background?

• P12, paragraph starting “previous studies showed…” should reference those previous studies.

• In legends for Figures 2A, 4C , 6C, Suppl Figs 4C, 7C: capitalize first word.

• Scalebars should be “Scale bars”

• GFP-COSA-1 should be GFP::COSA-1

• Suppl. Fig 7B, left word should read “damage” not “damages”

Methods:

Rather than referencing previous papers, please detail the methods for immunostaining, mass spectrometry, and fluorescence polarization (and define Scrum and Eurofin).

Describe how GFP-1::COSA-1 is visualized (antibody?)

PLOS authors have the option to publish the peer review history of their article (what does this mean?). If published, this will include your full peer review and any attached files.

Reviewer #1: No

Reviewer #2: No

Reviewer #3: No

**Have all data underlying the figures and results presented in the manuscript been provided?**

Reviewer #1: None

---

## [Decision Letter · Decision Letter 1]

8 Oct 2020

Dear Dr Carlton,

Thank you very much for submitting your Research Article entitled 'Phosphoregulation of HORMA domain protein HIM-3 promotes asymmetric synaptonemal complex disassembly in meiotic prophase in C. elegans' to PLOS Genetics. Your manuscript was fully evaluated at the editorial level and by independent peer reviewers. The reviewers appreciated the attention to an important topic but identified some minor aspects of the manuscript that should be improved by modifying the text.

We therefore ask you to modify the manuscript according to the review recommendations before we can consider your manuscript for acceptance. Your revisions should address the specific points made by each reviewer.

[LINK]

Yours sincerely,

Sarit Smolikove

Guest Editor

PLOS Genetics

Gregory P. Copenhaver

Editor-in-Chief

PLOS Genetics

There are few text edits suggested by reviewer 2 and 3 (see below) and I have few small edit suggestions indicated here:

Line 109: put a dot after [16-18]

Line 320: please insert a sentence telling the reader who is CHK-2

Ref 37 is now published and can replace the bioRxiv citation

Page 10-11: Figure 4C is discussed before 4B, so they should be switched

Page 12: Figure 5C is discussed before 5B, so they should be switched

Figure 1B: consider indicating the closer motifs of HTP-1/2 by a box the same way it’s done for HTP-3 and HIM-3

Figure 4 and 5 and 6D: on my computer it’s hard to see the magenta over the white background, consider moving the labels out of the panel to the top of each panel

Sup figure 5C, 7A, 13B, 14B: instead N2 please put “N2 wild type” or “wild type”

Figure Sup 7B: the word “Merged” is missing on some of the panels

Sup figure 9, 10, 13, 14C: instead control indicate/only indicate the specific genotype used

Reviewer's Responses to Questions

**Comments to the Authors:**

Reviewer #1: The authors have added a substantial amount of new data, which addressed some of the previous concerns. Especially, the distinct behavior of HIM-3 phosphorylation at S277 vs. S282 is intriguing, and identification of the kinases/phosphatases that are responsible for HIM-3 phosphorylation/dephosphorylation in synapsis- and crossover-dependent manners will be important topics for future research. Although the authors were not able to test the redundancy for establishing the asymmetric localization of HTP-1/2 due to technical difficulties, the additional experiment has placed HIM-3 phosphorylation downstream of PLK-2 and SYP-1 phosphorylation, which is a significant addition. Finally, the model presented in the supplemental figure is informative, and the revised Discussion adequately clarifies and explains the differential phenotypes in phospho-defective vs. phospho-mimetic mutants.

Overall, this study is well executed and will be of interest to the meiosis community. I support the publication of this work.

Reviewer #2: The revised manuscript by Sato-Carlton et al., describes the identification and functional characterization of HORMA axis protein, HIM-3 phosphorylation, in C. elegans meiosis. The authors have done a very good job addressing the previous reviews. I found a couple of typos:

1. Line 226 – please remove the first “single”

2. Line 516 – “remains” should be “remain”

Reviewer #3: The revised manuscript of Carlton et al addresses most of the comments of all of the reviewers and importantly adds statistics, new supplemental data, and important controls about the specificity of their antibody. While they are still lacking insight into the kinases and phosphatases that are controlling this modification, the manuscript as is highlights important biology that raises new areas for investigation that will be relevant to the field as a whole.

Three minor issues should be addressed for publication:

p. 1 last line: “…division of the chromosome into two segments or arms by..” I had to read this 3 times ot parse if correctly perhaps: “division of the chromosome into segments (or “arms”) by…”

Line 82: Although the prevailing dogma is that chromosomes receive ”one and only one crossover” there is actually genetics data from Hodgkin et al. (1979), that rare DCOs do occur. In males, Zetka and Rose reported DCOs in wild type and rec-1 mutants. Lim et al 2008 showed that there can be up to 3% DCOs on Chr III in hermaphrodites and 1% in males using molecular markers spanning >96% of the genome. Tsai et al 2008 found DCOs on chr I using cytological markers. So, using these data together and extrapolating to the whole genome, while most receive only a single crossover, telomere-proximal crossovers likely do happen in up to 15% of meioses on at least 1 chromosome.

Line 420-425: Since the table shows a statistically significant loss of viability in the Htp-1::flag; him-3(S282E) mutant, this needs to at least be addressed in the text and not only in the response to the reviewers.

**Have all data underlying the figures and results presented in the manuscript been provided?**

Reviewer #1: None

Reviewer #2: Yes

Reviewer #3: Yes

PLOS authors have the option to publish the peer review history of their article (what does this mean?). If published, this will include your full peer review and any attached files.

Reviewer #1: No

Reviewer #2: No

Reviewer #3: No

---

## [Editor Report · Decision Letter 2]

17 Oct 2020

Dear Dr Carlton,

We are pleased to inform you that your manuscript entitled "Phosphoregulation of HORMA domain protein HIM-3 promotes asymmetric synaptonemal complex disassembly in meiotic prophase in C. elegans" has been editorially accepted for publication in PLOS Genetics. Congratulations!

In preparing your final draft for the production team the editors ask that you address the following two issues:

1) Make sure this point is clearly addressed: "Line 420-425: Since the table shows a statistically significant loss of viability in the Htp-1::flag; him-3(S282E) mutant, this needs to at least be addressed

in the text and not only in the response to the reviewers." 

2) In Line 321 there is a problem with the references “(##references)” is placed instead of the actual ref number. The correct reference should be inserted with the right formatting.

Yours sincerely,

Sarit Smolikove

Guest Editor

PLOS Genetics

Gregory P. Copenhaver

Editor-in-Chief

PLOS Genetics

Comments from the reviewers (if applicable):

**Data Deposition**

http://datadryad.org/submit?journalID=pgenetics&manu=PGENETICS-D-20-01005R2

**Press Queries**

---

## [Editor Report · Acceptance letter]

27 Oct 2020

PGENETICS-D-20-01005R2 

Phosphoregulation of HORMA domain protein HIM-3 promotes asymmetric synaptonemal complex disassembly in meiotic prophase in *Caenorhabditis elegans*

Dear Dr Carlton, 

We are pleased to inform you that your manuscript entitled "Phosphoregulation of HORMA domain protein HIM-3 promotes asymmetric synaptonemal complex disassembly in meiotic prophase in *Caenorhabditis elegans*" has been formally accepted for publication in PLOS Genetics! Your manuscript is now with our production department and you will be notified of the publication date in due course.

With kind regards,

Matt Lyles

PLOS Genetics

On behalf of:
